# Parameterized Knowledge Transfer for Personalized Federated Learning

**Jie Zhang**[1], **Song Guo**[1,*], **Xiaosong Ma**[1], **Haozhao Wang**[2], **Wencao Xu**[1], and **Feijie Wu**[1]

[1]Department of Computing, The Hong Kong Polytechnic University
[2]Department of Computer Science and Technology, HUST
{jieaa.zhang,harli.wu}@connect.polyu.hk, hz_wang@hust.edu.cn,
{song.guo,xiaosma,wenchao.xu}@polyu.edu.hk

## Abstract

In recent years, personalized federated learning (pFL) has attracted increasing attention for its potential in dealing with statistical heterogeneity among clients. However, the state-of-the-art pFL methods rely on model parameters aggregation at the server side, which require all models to have the same structure and size, and thus limits the application for more heterogeneous scenarios. To deal with such model constraints, we exploit the potentials of heterogeneous model settings and propose a novel training framework to employ personalized models for different clients. Specifically, we formulate the aggregation procedure in original pFL into a personalized group knowledge transfer training algorithm, namely, KT-pFL, which enables each client to maintain a personalized soft prediction at the server side to guide the others' local training. KT-pFL updates the personalized soft prediction of each client by a linear combination of all local soft predictions using a knowledge coefficient matrix, which can adaptively reinforce the collaboration among clients who own similar data distribution. Furthermore, to quantify the contributions of each client to others' personalized training, the knowledge coefficient matrix is parameterized so that it can be trained simultaneously with the models. The knowledge coefficient matrix and the model parameters are alternatively updated in each round following the gradient descent way. Extensive experiments on various datasets (EMNIST, Fashion_MNIST, CIFAR-10) are conducted under different settings (heterogeneous models and data distributions). It is demonstrated that the proposed framework is the first federated learning paradigm that realizes personalized model training via parameterized group knowledge transfer while achieving significant performance gain comparing with state-of-the-art algorithms.

## 1 Introduction

Federated Learning (FL) [1] has emerged as an efficient paradigm to collaboratively train a shared machine learning model among multiple clients without directly accessing their private data. By periodically aggregating parameters from the clients for global model updating, it can converge to high accuracy and strong generalization. FL has shown its capability to protect user privacy while there remains a crucial challenge that significantly degrades the learning performance, i.e., statistic heterogeneity in users' local datasets. Given the Non-Independent and Identically Distributed (Non-IID) user data, the trained global model often cannot be generalized well over each client [2–5].

To deal with the above issues, employing personalized models appears to be an effective solution in FL, i.e., personalized federated learning (pFL). Recent works regarding pFL include regularization-based methods [6–8] (i.e., pFedMe [6], L2SGD [7], FedAMP [8]), meta-learning-based Per-FedAvg

---
*Corresponding author

35th Conference on Neural Information Processing Systems (NeurIPS 2021).

[9] and cluster-based IFCA [10, 11]. However, in order to aggregate the parameters from all clients, it is inevitable for them to have identical model structure and size. Such constraints would prevent status quo pFL methods from further application in practical scenarios, where clients are often willing to own unique models, i.e., with customized neural architectures to adapt to heterogeneous capacities in computation, communication and storage space, etc. Motivated by the paradigm of Knowledge Distillation (KD) [12–16] that knowledge can be transferred from a neural network to another via exchanging soft predictions instead of using the whole model parameters, KD-based FL training methods have been studied [12, 14–19] to collaboratively train heterogeneous models in a privacy-preserving way. However, these works have neglected the further personalization requirement of FL clients, which can be well satisfied via the personalized knowledge transfer for heterogeneous FL users.

In this paper, we seek to develop a novel training framework that can accommodate heterogeneous model structures for each client and achieve personalized knowledge transfer in each FL training round. To this end, we formulate the aggregation phase in FL to a personalized group knowledge transfer training algorithm dubbed KT-pFL, whose main idea is to allow each client to maintain a personalized soft prediction at the server that can be updated by a linear combination of all clients' local soft predictions using a knowledge coefficient matrix. The principle of doing so is to reinforce the collaboration between clients with similar data distributions. Furthermore, to quantify the contribution of each client to other's personalized soft prediction, we parameterize the knowledge coefficient matrix so that it can be trained simultaneously with the models following an alternating way in each iteration round.

We show that KT-pFL not only breaks down the barriers of homogeneous model restriction, which requires to transfer the entire parameters set in each round, whose data volume is much larger than that of the soft prediction, but also improves the training efficiency by using a parameterized update mechanism. Experimental results on different datasets and models show that our method can significantly improve the training efficiency and reduce the communication overhead. Our contributions are:

• To the best of our knowledge, this paper is the first to study the personalized knowledge transfer in FL. We propose a novel training framework, namely, KT-pFL, that maintains a personalized soft prediction for each client in the server to transfer knowledge among all clients.
• To encourage clients with similar data distribution to collaborate with each other during the training process, we propose the 'knowledge coefficient matrix' to identify the contribution from one client to others' local training. To show the efficiency of the parameterized method, we compared KT-pFL with two non-parameterized learning methods, i.e., TopK-pFL and Sim-pFL, which calculate the knowledge coefficient matrix on the cosine similarity between different model parameters.
• We provide theoretical performance guarantee for KT-pFL and conduct extensive experiments over various deep learning models and datasets. The efficiency superiority of KT-pFL is demonstrated by comparing our proposed training framework with traditional pFL methods.

## 2 Related Work

**Personalized FL with Homogeneous Models.** Recently, various approaches have been proposed to realize personalized FL with homogeneous local model structure, which can be categorized into three types according to the number of global models applied in the server, i.e., single global model, multiple global models and no global model.

single global model type is a close variety of conventional FL, e.g., FedAvg [1], that combine global model optimization process with additional local model customization, and consist of four different kinds of approaches: local fine-tuning [20–23], regularization (e.g., pFedMe [6], L2SGD [7, 24], Ditto [25]), hybrid local and global models [11, 26, 27] and meta learning [9, 28]. All of these pFL methods apply a single global model, and thus limit the customized level of the local model at the client side. Therefore, some researchers [8, 10, 11] propose to train multiple global models at the server, where clients are clustered into several groups according to their similarity and different models are trained for each group. FedAMP [8] and FedFomo [29] can be regarded as special cases of the clustered-based method that each client owns a personalized global model at the server side. As a contrast, some literature waive the global model to deal with the heterogeneity problem [30, 31], such as multi-task learning based (i.e., MOCHA [31]) and hypernetwork-based framework (i.e.,

FedHN [30]). However, all these methods require aggregating the model parameters from the clients, who have to apply identical model structure and size, which hinders further personalization, e.g., employing personalized model architectures for heterogeneous clients is not feasible.

**Heterogeneous FL and Knowledge Distillation.** To enable heterogeneous model architectures in FL, Diao et al. [32] propose to upload a different subset of global model to the server for aggregation with the objective to produce a single global inference model. Another way of personalization is to use Knowledge Distillation (KD) in heterogeneous FL systems ([12, 14, 15, 17–19, 33, 34]). The principle is to aggregate local soft-predictions instead of local model parameters in the server, whereby each client can update the local model to approach the averaged global predictions. As KD is independent with model structure, some literature [13, 16] are proposed to take advantage of such independence to implement personalized FL with heterogeneous models at client sides. For example, Li et al. [13] propose FedMD to perform ensemble distillation for each client to learn well-personalized models. Different from FedMD that exchanging soft-predictions between the clients and the server, FedDF [16] first aggregates local model parameters for model averaging at the server side. Then, the averaged global models can be updated by performing knowledge transfer from all received (heterogeneous) client models.

To sum up, most of these schemes construct an ensembling teacher by simply averaging the teachers' soft predictions, or by heuristically combining the output of the teacher models, which are far away from producing optimal combination of teachers. In our framework, KD is used in a more efficient way that the weights of the clients' soft predictions are updated together with the model parameters during every FL training iteration.

## 3 Problem Formulation

We aim to collaboratively train personalized models for a set of clients applying different model structures in FL. Consider supervised learning whose goal is to learn a function that maps every input data to the correct class out of $\mathcal{C}$ possible options. We assume that there are $N$ clients, and each client $n$ can only access to his private dataset $\mathbb{D}_n := \{x_i^n, y_i\}$, where $x_i$ is the $i$-th input data sample, $y_i$ is the corresponding label of $x_i$, $y_i \in \{1, 2, \cdots, C\}$. The number of data samples in dataset $\mathbb{D}_n$ is denoted by $D_n$. $\mathbb{D} = \{\mathbb{D}_1, \mathbb{D}_2, \cdots, \mathbb{D}_N\}$, $D = \sum_{n=1}^{N} D_n$. In conventional FL, the goal of the learning system is to learn a global model $\mathbf{w}$ that minimizes the total empirical loss over the entire dataset $\mathbb{D}$:

$$\min_{\mathbf{w}} \mathcal{L}(\mathbf{w}) := \sum_{n=1}^{N} \frac{D_n}{D} \mathcal{L}_n(\mathbf{w}), \text{ where } \mathcal{L}_n(\mathbf{w}) = \frac{1}{D_n} \sum_{i=1}^{D_n} \mathcal{L}_{CE}(\mathbf{w}; x_i, y_i), \tag{1}$$

where $\mathcal{L}_n(\mathbf{w})$ is the $n$-th client's local loss function that measures the local empirical risk over the private dataset $\mathbb{D}_n$ and $\mathcal{L}_{CE}$ is the cross-entropy loss function that measures the difference between the predicted values and the ground truth labels.

However, this formulation requires all local clients to have a unified model structure, which cannot be extended to more general cases where each client applies a unique model. Therefore, we need to reformulate the above optimization problem to break the barrier of homogeneous model structure. Besides, given the Non-IID clients' datasets, it is inappropriate to just minimize the total empirical loss (i.e., $\min_{\mathbf{w}^1, \cdots, \mathbf{w}^N} \mathcal{L}(\mathbf{w}^1, \cdots, \mathbf{w}^N) := \sum_{n=1}^{N} \frac{D_n}{D} L_n(\mathbf{w}^n)$). To that end, we propose the following training framework:

**Definition 3.1.** Let $s(\mathbf{w}^n, \hat{x})$ denote the *collaborative knowledge* from client $n$, and $\hat{x}$ denote a data sample from a public dataset $\mathbb{D}_r$ that all clients can access to. Define the personalized loss function of client $n$ as

$$\mathcal{L}_{per,n}(\mathbf{w}^n) := \mathcal{L}_n(\mathbf{w}^n) + \lambda \sum_{\hat{x} \in \mathbb{D}_r} \mathcal{L}_{KL} \left( \sum_{m=1}^{N} c_{mn} \cdot s(\mathbf{w}^m, \hat{x}), s(\mathbf{w}^n, \hat{x}) \right), \tag{2}$$

where $\lambda > 0$ is a hyper-parameters, $\mathcal{L}_{KL}$ stands for Kullback–Leibler (KL) Divergence function and is added to the loss function to transfer personalized knowledge from a teacher to another. $c_{mn}$ is the knowledge coefficient which is used to estimate the contribution from client $m$ to $n$. The second term in (2) allows each client to build his own personalized aggregated knowledge in the server and enhance the collaboration effect between clients with large $\mathbf{c}$. The concept of *collaborative*

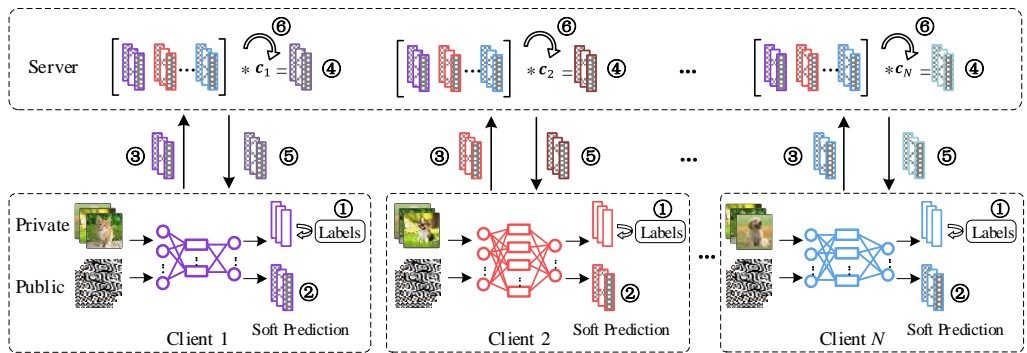

Figure 1: Illustration of the KT-pFL framework. The workflow includes 6 steps: ① local training on private data; ②, ③ each client outputs the local soft prediction on public data and sends it to the server; ④ the server calculates each client's personalized soft prediction via a linear combination of local soft predictions and knowledge coefficient matrix; ⑤ each client downloads the personalized soft prediction to perform distillation phase; ⑥ the server updates the knowledge coefficient matrix.

*knowledge* can either refer to soft predictions or model parameters depending on the definition in different situations. For example, $s(\mathbf{w}^n, \hat{x})$ can be deemed to be a soft prediction of the client $n$, which are calculated with the softmax of logits $z^n$, i.e., $s(\mathbf{w}^n, \hat{x}) = \frac{\exp(z_c^n/T)}{\sum_{c=1}^{C} \exp(z_c^n/T)}$, logits $z^n$ is the output of the last fully connected layer on client $n$'s model, $T$ is the temperature hyperparameter of the softmax function.

**Definition 3.2.** We define $\mathbf{c} \in \mathbb{R}^{N \times N}$ as the *knowledge coefficient matrix*:

$$\mathbf{c} = \begin{Bmatrix} c_{11} & c_{12} & \cdots & c_{1N} \\ c_{21} & c_{22} & \cdots & c_{2N} \\ \vdots & \vdots & \ddots & \vdots \\ c_{N1} & c_{N2} & \cdots & c_{NN} \end{Bmatrix}. \tag{3}$$

Our objective is to minimize

$$\min_{\mathbf{w}, \mathbf{c}} \mathcal{L}(\mathbf{w}, \mathbf{c}) := \sum_{n=1}^{N} \frac{D_n}{D} \mathcal{L}_{per,n}(\mathbf{w}^n) + \rho \|\mathbf{c} - \frac{\mathbf{1}}{N}\|^2, \tag{4}$$

where $\mathbf{w} = [\mathbf{w}^1, \cdots, \mathbf{w}^N] \in \mathbb{R}^{\sum_{n=1}^{N} d_n}$ is the concatenated vector of all weights, $d_n$ represents the dimensions of model parameter $\mathbf{w}^n$. $\mathbf{1} \in \mathbb{R}^{n^2}$ is the identity matrix whose elements are all equal to 1. The second term in (4) is a regularization term that ensures generalization ability of the whole learning system. Without the regularization term, a client with a completely different data distribution tends to set large values of the knowledge coefficient (i.e., equal to 1), and no collaboration will be conducted during training in this case. $\rho$ is a regularization parameter that is larger than 0.

# 4   KT-pFL Algorithm

In this section, we introduce the proposed KT-pFL algorithm, where the local model parameters and knowledge coefficient matrix are updated alternatively. To enable personalized knowledge transfer in FL, we train personalized models locally according to the related *collaborative knowledge*. Insert (2) into (4), and we can design an alternating optimization approach to solve (4), that in each round we fix either $\mathbf{w}$ or $\mathbf{c}$ by turns, and optimize the unfixed one following an alternating way until a convergence point is reached.

**Update $\mathbf{w}$:** In each communication round, we first fix $\mathbf{c}$ and optimize (train) $\mathbf{w}$ for several epochs locally. In this case, updating $\mathbf{w}$ depends on both the private data (i.e., $\mathcal{L}_{CE}$ on $\mathbb{D}_n, n \in [1, \cdots, N]$), that can only be accessed by the corresponding client, and the public data (i.e., $\mathcal{L}_{KL}$ on $\mathbb{D}_r$), which is accessible for all clients. We propose a two-stage updating framework for $\mathbf{w}$:

---

**Algorithm 1** KT-pFL Algorithm

---
**Input:** $\mathbb{D}, \mathbb{D}_r, \eta_1, \eta_2, \eta_3$ and $T$
**Output:** $\mathbf{w} = [\mathbf{w}^1, \cdots, \mathbf{w}^N]$
 1: Initialize $\mathbf{w}_0$ and $\mathbf{c}_0$
 2: **procedure** SERVER-SIDE OPTIMIZATION
 3:     Distribute $\mathbf{w}_0$ and $\mathbf{c}_0$ to each client
 4:     **for** each communication round $t \in \{1, 2, ..., T\}$ **do**
 5:         **for** each client $n$ **in parallel do**
 6:             $\mathbf{w}_{t+1}^n \leftarrow ClientLocalUpdate(n, \mathbf{w}_t^n, \mathbf{c}_{t,n})$
 7:         Update knowledge coefficient matrix $\mathbf{c}$ via (7)
 8:         Distribute $\mathbf{c}_{t+1}$ to all clients
 9: **procedure** CLIENTLOCALUPDATE($n, \mathbf{w}_t^n, \mathbf{c}_{t,n}$)
10:     Client $n$ receives $\mathbf{w}_t^n$ and $\mathbf{c}_n$ from the server
11:     **for** each local epoch $i$ from 1 to $E$ **do**
12:         **for** mini-batch $\xi_t \subseteq \mathbb{D}_n$ **do**
13:             **Local Training:** update model parameters on private data via (5)
14:     **for** each distillation step $j$ from 1 to $R$ **do**
15:         **for** mini-batch $\xi_{r,t} \subseteq \mathbb{D}_r$ **do**
16:             **Distillation:** update model parameters on public data via (6)
        **return** local parameters $\mathbf{w}_{t+1}^n$

---

- *Local Training*: Train $\mathbf{w}$ on each client's private data by applying a gradient descent step:

$$\mathbf{w}^n \leftarrow \mathbf{w}^n - \eta_1 \nabla_{\mathbf{w}^n} \mathcal{L}_n(\mathbf{w}^n; \xi_n), \tag{5}$$

  where $\xi_n$ denotes the mini-batch of data $\mathbb{D}_n$ used in local training, $\eta_1$ is the learning rate.

- *Distillation*: Transfer knowledge from personalized soft prediction to each local client based on public dataset:

$$\mathbf{w}^n \leftarrow \mathbf{w}^n - \eta_2 \nabla_{\mathbf{w}^n} \mathcal{L}_{KL} \left( \sum_{m=1}^N \mathbf{c}_m^{*,T} \cdot s(\mathbf{w}^m, \xi_r), s(\mathbf{w}^n, \xi_r) \right), \tag{6}$$

  where $\xi_r$ denotes the mini-batch of public data $\mathbb{D}_r$, and $\eta_2$ is the learning rate. $\mathbf{c}_m^* = [c_{m1}, c_{m2}, \cdots, c_{mN}]$ is the *knowledge coefficient* vector for client $m$, which can be found in $m$-th row of $\mathbf{c}$. Note that all *collaborative knowledge* and *knowledge coefficient matrix* are required to obtain the personalized soft prediction in this stage, which can be collected in the server.

**Update c:** After updating $\mathbf{w}$ locally for several epochs, we turn to fix $\mathbf{w}$ and update $\mathbf{c}$ in the server.

$$\mathbf{c} \leftarrow \mathbf{c} - \eta_3 \lambda \sum_{n=1}^N \frac{D_n}{D} \nabla_{\mathbf{c}} \mathcal{L}_{KL} \left( \sum_{m=1}^N \mathbf{c}_m \cdot s(\mathbf{w}^{m,*}, \xi_r), s(\mathbf{w}^{n,*}, \xi_r) \right) - 2\eta_3 \rho (\mathbf{c} - \frac{1}{N}), \tag{7}$$

where $\eta_3$ is the learning rate for updating $\mathbf{c}$.

Algorithm 1 demonstrates the proposed KT-pFL algorithm and the idea behind it is shown in Figure 1. In every communication round of training, the clients use local SGD to train several epochs based on the private data and then send the *collaborative knowledge* (e.g., soft predictions on public data) to the server. When the server receives the *collaborative knowledge* from each client, it aggregates them to form the personalized soft predictions according to the knowledge coefficient matrix. The server then sends back the personalized soft prediction to each client to perform local distillation. The clients then iterate for multiple steps over public dataset [2]. After that, the knowledge coefficient matrix is updated in the server while fixing the model parameters $\mathbf{w}$.

**Performance Guarantee.** Theorem 4.1 provides the performance analysis of the personalized model when each client owns Non-IID data. Detailed description and derivations are deferred to Appendix A.

**Theorem 4.1.** *Denote the $n$-th local distribution and its empirical distribution by $\mathcal{D}_n$ and $\hat{\mathcal{D}}_n$ respectively, and the hypothesis $h \in \mathcal{H}$ trained on $\hat{\mathcal{D}}_n$ by $h_{\hat{\mathcal{D}}_n}$. There always exist $c_{m,n}^*, m = 1, \ldots, N,$*

---
[2]Note that our method can work over both labeled and unlabeled public datasets.

*such that the expected loss of the personalized ensemble model for the data distribution $\mathcal{D}_n$ of client $n$ is not larger than that of the single model only trained with local data: $\mathcal{L}_{\mathcal{D}_n}(\sum_{m=1}^{N} c_{m,n}^* h_{\hat{\mathcal{D}}_m}) \leq \mathcal{L}_{\mathcal{D}_n}(h_{\hat{\mathcal{D}}_n})$. Besides, there exist some problems where the personalized ensemble model is strictly better, i.e., $\mathcal{L}_{\mathcal{D}_n}(\sum_{m=1}^{N} c_{m,n}^* h_{\hat{\mathcal{D}}_m}) < \mathcal{L}_{\mathcal{D}_n}(h_{\hat{\mathcal{D}}_n})$.*

Theorem 4.1 indicates that the performance of the personalized ensemble model under some suitable coefficient matrices is better than that of the model only trained on its local private data, which theoretically demonstrates the necessity of the parameterized personalization. However, it is challenging to find such matrices due to the complexity and diversity of machine learning models and data distributions. In this paper, our designed algorithm KT-pFL can find the desired coefficient matrix in a gradient descent manner, hence achieving the performance boost. Besides, we have the similar claim for the relationship between the personalized ensemble model and average ensemble model.

**Remark 4.1.** *There always exist $c_{m,n}^*, m = 1, \ldots, N$, such that the expected loss of the personalized ensemble model for the data distribution $\mathcal{D}_n$ of client $n$ is not larger than that of the average ensemble model: $\mathcal{L}_{\mathcal{D}_n}(\sum_{m=1}^{N} c_{m,n}^* h_{\hat{\mathcal{D}}_m}) \leq \mathcal{L}_{\mathcal{D}_n}(\frac{1}{N} \sum_{m=1}^{N} h_{\hat{\mathcal{D}}_m})$. Besides, there exist some problems where the personalized ensemble model is strictly better, i.e., $\mathcal{L}_{\mathcal{D}_n}(\sum_{m=1}^{N} c_{m,n}^* h_{\hat{\mathcal{D}}_m}) < \mathcal{L}_{\mathcal{D}_n}(\frac{1}{N} \sum_{m=1}^{N} h_{\hat{\mathcal{D}}_m})$.*

## 5 Evaluations

### 5.1 Experimental Setup

**Task and Datasets** We evaluate our proposed training framework on three different image classification tasks: EMNIST [35], Fashion_MNIST [36] and CIFAR-10 [37]. For each dataset, we apply two different Non-IID data settings: 1) each client only contains two classes of samples; 2) each client contains all classes of samples, while the number of samples for each class is different from that of a different client. All datasets are split randomly with 75% and 25% for training and testing, respectively. The testing data on each client has the same distribution with its training data. For all methods, we record the average test accuracy of all local models for evaluation.

**Model Structure:** Four different lightweight model structures including LeNet [38], AlexNet [39], ResNet-18 [40], and ShuffleNetV2 [41] are adopted in our experiments. Our pFL system has 20 clients, who are assigned with four different model structures, i.e., five clients per model.

**Baselines:** Although KT-pFL is designed for effective personalized federated learning, it can be applied to both heterogeneous and homogeneous model cases. We first conduct experiments on heterogeneous systems where the neural architecture of the local models are different among clients. We compare the performance of KT-pFL to the non-personalized distillation-based methods: FedMD [13], FedDF [16] and the personalized distillation-based method pFedDF[3] and other simple versions of KT-pFL including Sim-pFL and TopK-pFL. Sim-pFL calculates the knowledge coefficient by using cosine similarity between two local soft predictions. Instead of combining the knowledge from all clients, TopK-pFL obtains the personalized soft predictions only from $K$ clients[4] who have higher value of cosine similarity.

To demonstrate the generalization and effectiveness of our proposed training framework, we further compare KT-pFL to FedAvg [1] and state-of-the-art pFL methods including Per-Fedavg [9], Fedavg-Local FT[23], pFedMe [6], FedAMP [8], FedFomo[29] and FedHN[30] under the homogeneous model setting. Note that Per-FedAvg is a MAML-based method which aims to optimize the one-step gradient update for its personalized model. pFedMe and FedAMP are two regularized-based methods. The former one obtains personalized models by controlling the distance between local model and global model, while the later one facilitates the collaboration of clients with similar data distribution.

**Implementation** The experiments are implemented in PyTorch. We simulate a set of clients and a centralized server on one deep learning workstation (i.e., Intel(R) Core(TM) i9-9900KF CPU@3.6GHz with one NVIDIA GeForce RTX 2080Ti GPU).

---

[3]We use the fused prototype models at the last round of FedDF as the pFedD's initial models. Each client fine-tunes it on the local private dataset with several epochs.

[4]In our experiments, we set $K$ as 5.

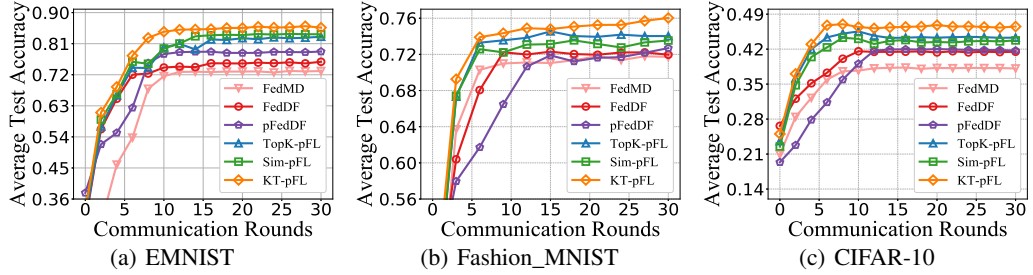

Figure 2: Performance comparison of FedMD, FedDF, pFedDF, TopK-pFD, Sim-pFD, and KT-pFL in average test accuracy on three datasets (Non-IID case 1: each client contains all labels).

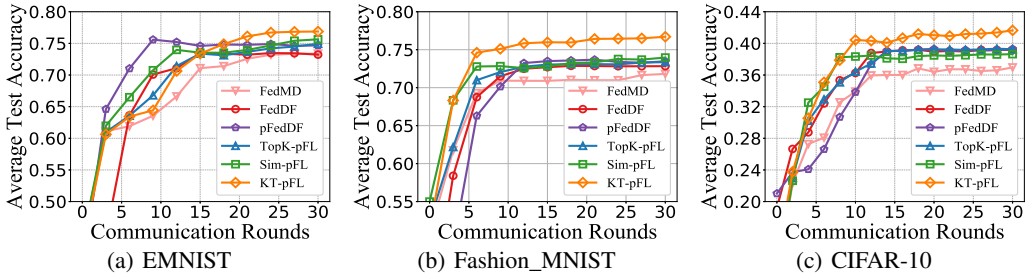

Figure 3: Performance comparison of FedMD, FedDF, pFedDF, TopK-pFD, Sim-pFD, and KT-pFL in average test accuracy on three datasets (Non-IID case 2: each client contains only two labels).

## 5.2 Results

Subject to space constraints, we only report the most important experimental results in this section. Please refer to Appendix B for the details of different Non-IID data settings on EMNIST, Fashion_MNIST and CIFAR10, the implementation details and the hyperparameter settings of all the methods, and also extra results about the convergence and robustness analysis. In our experiments, we run each experiment multiple times and record the average results.

Table 1: The comparison of final test accuracy (%) on different datasets with heterogeneous models (i.e., Lenet, AlexNet, ResNet-18 and ShuffleNetV2).

| Method | EMNIST | | Fashion_MNIST | | CIFAR-10 | |
|---|---|---|---|---|---|---|
| | Non-IID_1 | Non-IID_2 | Non-IID_1 | Non-IID_2 | Non-IID_1 | Non-IID_2 |
| FedMD [13] | 71.37 | 70.15 | 68.02 | 66.10 | 39.14 | 35.56 |
| FedDF [16] | 71.92 | 72.79 | 70.32 | 70.83 | 40.19 | 37.81 |
| pFedDF | 78.59 | 75.72 | 72.69 | 73.45 | 42.83 | 39.54 |
| Sim-pFD | 82.72 | 74.42 | 72.82 | 72.71 | 42.50 | 39.90 |
| TopK-pFD | 83.00 | 73.91 | 72.24 | 73.39 | 42.58 | 39.34 |
| KT-pFL | **84.65** | **76.30** | **75.48** | **75.60** | **43.82** | **41.04** |

### 5.2.1 Performance Comparison

**Heterogeneous FL.** For all the methods and all the data settings, the batch size on private data and public data are 128 and 256, respectively, the number of local epochs is 20 and the distillation steps is 1 in each communication round of pFL training. Unless mentioned, otherwise the public data used for EMNIST and Fashion_MNIST is MNIST, and the public data used for CIFAR-10 is CIFAR-100. the size of public data used in each communication round is 3000, the learning rate is set to 0.01 for EMNIST and Fashion_MNIST, and 0.02 for CIFAR-10.

Figure 2 and 3 show the curves of the average test accuracy during the training process on four different models with three datasets, which include the results of FedMD, FedDF, pFedDF, Sim-pFL,

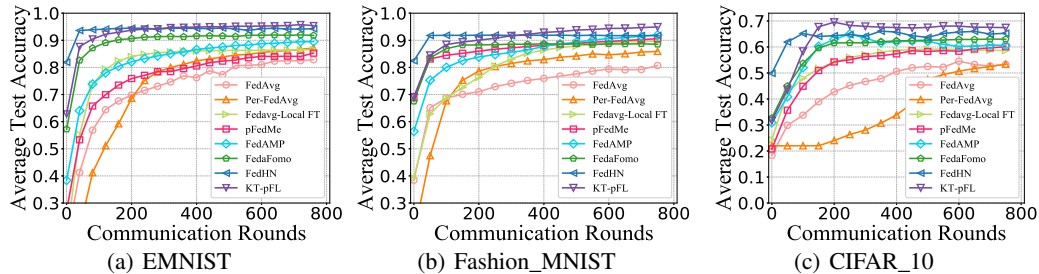

|  | (a) EMNIST | (b) Fashion_MNIST | (c) CIFAR_10 |

Figure 4: Performance comparison of FedAvg, Per-Fedavg, Fedavg-Local FT, pFedMe, FedAMP, FedFomo, FedHN and KT-pFL in average test accuracy on three datasets. The Non-IID data setting: each client contains all labels. 20 clients with homogeneous models: CNN [1]. Learning rate: 0.005.

Table 2: The comparison of final test accuracy (%) on different datasets with different number of homogeneous models (i.e., the same CNN architecture as [1]). For large-scale FL system with 100 clients, we set the client sampling rate as 0.1.

| Method | EMNIST | | Fashion_MNIST | | CIFAR-10 | |
|---|---|---|---|---|---|---|
| | 20 clients | 100 clients | 20 clients | 100 clients | 20 clients | 100 clients |
| Fedavg [1] | 85.86 | 76.84 | 80.69 | 72.36 | 51.94 | 43.56 |
| Per-Fedavg [9] | 86.16 | 82.87 | 85.91 | 77.93 | 55.61 | 49.82 |
| Fedavg-Local FT [23] | 86.59 | 84.55 | 90.17 | 80.93 | 42.88 | 50.91 |
| pFedMe [6] | 84.26 | 86.14 | 90.97 | 84.02 | 62.79 | 52.46 |
| FedAMP [8] | 88.51 | 85.63 | 90.79 | 84.72 | 60.75 | 51.26 |
| FedFomo [29] | 92.58 | 90.08 | 88.33 | 87.76 | 63.11 | 55.73 |
| pFedHN [30] | 94.27 | 92.21 | 92.22 | 89.14 | 65.31 | 57.38 |
| KT-pFL | **94.40** | **92.50** | **93.93** | **90.37** | **66.96** | **58.29** |

TopK-pFL and KT-pFL. We also summarize the final average test accuracy in Table 1. In two cases of Non-IID settings, KT-pFL obtains comparable or even better accuracy performance than others. The performances of FedMD, FedDF, pFedDF are worse than the other three methods. The reason is that taking the global aggregation of all local soft predictions trained on the Non-IID data from different clients produces only one global soft prediction, which cannot be well adapted to each client. The other two personalized methods, i.e., Sim-pFL and TopK-pFL, achieve comparably performance on most of cases with pFedDF. Our proposed method KT-pFL has the best performance, because each client can adaptively aggregate all local soft predictions to form a personalized one instead of being restricted to a global soft prediction.

**Homogeneous FL.** Apart from heterogeneous system, we further extend KT-pFL to homogeneous model setting. To conduct fair comparison with baselines, we exchange model parameters with the server to perform knowledge transfer. Specifically, each client maintains a personalized global model at the server side and each personalized global model is aggregated by a linear combination of local model parameters and knowledge coefficient matrix (Appendix C). In this case, we compare the performance of homogeneous-version of KT-pFL with FedAvg, Per-Fedavg, Fedavg-Local FT, pFedMe, FedAMP, FedFomo and FedHN. Figure 4 and Table 2 show the curve of the average test accuracy during training and the final average test accuracy on three different datasets, respectively. For all datasets, KT-pFL dominates the other seven methods on average test accuracy with less variance. Besides, the concept of FedAMP and FedFomo is similar with our proposed method KT-pFL. The difference is that the weights for each local model during personalized aggregation phase are updated in a gradient descent manner in KT-pFL.

To verify the efficiency of KT-pFL on large-scale FL system, we conduct experiments on 100 clients and apply the same client selection mechanism as other baselines. Specifically, in KT-pFL, only partial of elements in the knowledge coefficient matrix **c** will be updated in each communication round on large-scale FL systems. From the results in Table 2, it is obvious that our proposed method can work well both on small-scale and large-scale FL systems.

Table 3: The comparison of final test accuracy on different setting of local epochs $E$ and distillation steps $R$.

| Dataset | # Local epochs ($E$) | | | | # Distillation steps ($R$) | | | |
|---|---|---|---|---|---|---|---|---|
| | 5 | 10 | 15 | 20 | 1 | 2 | 3 | 5 |
| EMNIST (%) | 90.15 | **92.76** | 91.03 | 90.15 | **91.76** | 91.40 | 91.18 | 91.54 |
| Fashion_MNIST (%) | 88.73 | 89.14 | 88.58 | **89.42** | 89.14 | **89.90** | 89.07 | 88.08 |
| CIFAR-10 (%) | 58.30 | **59.38** | 59.34 | 59.24 | **59.24** | 57.99 | 59.22 | 58.91 |

Table 4: The comparison of final test accuracy on different setting of regularization parameter $\rho$.

| Dataset | # Regularization parameter $\rho$ | | | |
|---|---|---|---|---|
| | 0.1 | 0.3 | 0.5 | 0.7 |
| EMNIST (%) | 90.96 | 90.66 | 90.88 | **91.76** |
| Fashion_MNIST (%) | 89.49 | 89.30 | **89.56** | 89.14 |
| CIFAR-10 (%) | 59.08 | 58.52 | **59.56** | 58.40 |

#### 5.2.2 Effect of hyperparameters

To understand how different hyperparameters such as $E$, $R$, $\rho$, and $|\mathbb{D}_r|$ can affect the training performance of KT-pFL in different settings, we conduct various experiments on three datasets with $\eta_1 = \eta_2 = \eta_3 = 0.01$.

**Effect of Local Epochs $E$.** To reduce communication overhead, the server tends to allow clients to have more local computation steps, which can lead to less global updates and thus faster convergence. Therefore, we monitor the behavior of KT-pFL using a number of values of $E$, whose results are given in Table 3. The results show that larger values of $E$ can benefit the convergence of the personalized models. There is, nevertheless, a trade-off between the computations and communications, i.e., while larger $E$ requires more computations at local clients, smaller $E$ needs more global communication rounds to converge. To do such trade-off, we fix $E = 20$ and evaluate the effect of other hyperparameters accordingly.

**Effect of Distillation Steps $R$.** As the performance of the knowledge transfer is directly related to the number of distillation steps $R$, we compare the performance of KT-pFL under different setting of distillation steps (e.g., $R = 1, 2, 3, 5$). The number of local epochs is set to 20. The results show that larger value of $R$ cannot always lead to better performance, which means a moderate number of the distillation steps is necessary to approach to the optimal performance.

**Effect of $\rho$.** As mentioned in Eq (4), $\rho$ is the regularization term to do the trade-off between personalization ability and generalization ability. For example, larger value of $\rho$ means that the knowledge coefficient of each local soft prediction should approach to $\frac{1}{N}$, and thus more generalization ability should be guaranteed. Table 4 shows the results of KT-pFL with different value of $\rho$. In most settings, a significantly large value of $\rho$ will hurt the performance of KT-pFL.

Table 5: Performance of KT-pFL under different setting of public dataset (Task: Fashion_MNIST; Unlabeled Open Dataset: divided from Fashion_MNIST).

| Public dataset | Test Accuracy |
|---|---|
| MNIST | 89.14 ($\pm$ 0.15) |
| EMNIST | 88.76 ($\pm$ 0.21) |
| Unlabeled Open Dataset | 89.03 ($\pm$ 0.11) |

**Effect of different public dataset.** Besides, we compare the performance of our proposed method on different public datasets (Table 5). From the experimental results, we can observe that different settings of the public dataset have little effect on the training performance.

**Effect of $|\mathbb{D}_r|$.** Table 6 demonstrates the effect of public data size $|\mathbb{D}_r|$ used in local distillation phase. When the size of the public data is increased, KT-pFL has higher average test accuracy. However,

Table 6: The comparison of final test accuracy on different setting of the public data size $|\mathbb{D}_r|$.

| Dataset | # Size of public data ($|\mathbb{D}_r|$) | | | | | |
|---|---|---|---|---|---|---|
| | 10 | 100 | 500 | 1000 | 3000 | 5000 |
| EMNIST (%) | 56.19 | 73.44 | 91.25 | 91.47 | **91.66** | 91.32 |
| Fashion_MNIST (%) | 50.80 | 67.43 | 88.08 | 89.40 | **89.50** | 89.14 |
| CIFAR-10 (%) | 39.08 | 47.70 | **58.93** | 58.79 | 58.91 | 58.40 |

very large $|\mathbb{D}_r|$ will not only slow down the convergence of KT-pFL but also incur higher computation time at the clients. During the experiments, the value of $|\mathbb{D}_r|$ is configured to 3000.

### 5.2.3 Efficiency Evaluation

To evaluate the communication overhead, we record three aspects information: Data (batch size used in distillation phase), Param (model parameters) and Soft Prediction without using data compression techniques. The results are shown in Figure 5. Compared with conventional parameter-based pFL methods, the communication overhead on soft prediction-based KT-pFL is far less than Conv-pFL.

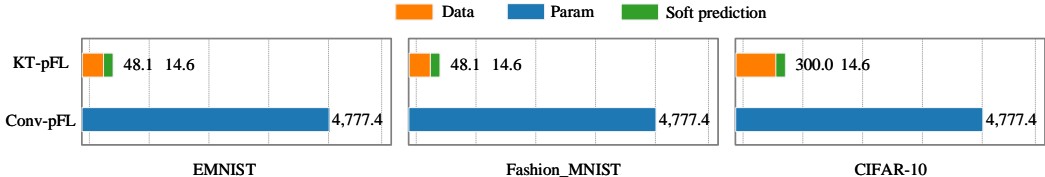

Figure 5: Communication Efficiency (X-axis units: MBytes) on three datasets. KL-pFL: soft prediction-based personalized federated learning; Conv-pFL: conventional parameter-based personalized federated learning. (20 models, 30 communication rounds for all datasets. In this experiment, the public dataset is stored on the server.)

## Broader Impact

FL has been emerged as new paradigm to collaboratively train models among multiple clients in a privacy-preserving manner. Due to the diversity of users (e.g., statistical and systematic heterogeneity, etc.), applying personalization in FL is essential for future trend. Our method KT-pFL not only breaks the barriers of homogeneous model constraint, which can significantly reduce the communication overhead during training, but also improves the training efficiency via a parameterized update mechanism without additional computation overhead at the client side. This research has the potential to enable various devices to cooperatively train ML tasks based on customized neural network architectures.

## Acknowledgements

This research was supported by the funding from Hong Kong RGC Research Impact Fund (RIF) with the Project No. R5060-19, General Research Fund (GRF) with the Project No. 152221/19E and 15220320/20E, the National Natural Science Foundation of China (61872310), and Shenzhen Science and Technology Innovation Commission (R2020A045).

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
