# A  Details on Performance Guarantee

In this appendix, we provide detailed performance analysis for the personalized model when each client owns Non-IID data. Firstly, we make some notations for our proof. In pFL, each client has access to its own data distribution $\mathcal{D}_n$ over domain $\Xi := \mathcal{X} \times \mathcal{Y}$, where $\mathcal{X} \in \mathbb{R}^d$ is the input space and $\mathcal{Y}$ is the output space. For our analysis, we assume a binary classification task (i.e., logistic regression), with hypothesis $h$ as a function $h : \mathcal{X} \to \{0, 1\}$ and $d = 2$. The loss function of the logistic regression is denoted by $\ell(h(x), y) = -y \log(\hat{y}) - (1 - y) \log(1 - \hat{y})$, where $\hat{y} := h_{\mathbf{w}}(x) = \frac{1}{1 + \exp -(\mathbf{w}^T x + b)}$. We denote $\arg \min_{h \in \mathcal{H}} \mathcal{L}_{\hat{D}}(h) = \frac{1}{|\hat{D}|} \sum_{(x,y) \in \hat{D}} \ell(h(x), y)$ by $h_{\hat{D}}$.

**Theorem A.1.** *Denote the $n$-th local distribution and its empirical distribution by $\mathcal{D}_n$ and $\hat{\mathcal{D}}_n$ respectively, and the hypothesis $h \in \mathcal{H}$ trained on $\hat{\mathcal{D}}_n$ by $h_{\hat{\mathcal{D}}_n}$. There always exist $c^*_{m,n}, m = 1, \dots, N$, such that the expected loss of the personalized ensemble model for the data distribution $\mathcal{D}_n$ of client $n$ is not larger than that of the single model only trained with local data: $\mathcal{L}_{\mathcal{D}_n}(\sum_{m=1}^N c^*_{m,n} h_{\hat{\mathcal{D}}_m}) \leq \mathcal{L}_{\mathcal{D}_n}(h_{\hat{\mathcal{D}}_n})$. Besides, there exist some problems where the personalized ensemble model is strictly better, i.e., $\mathcal{L}_{\mathcal{D}_n}(\sum_{m=1}^N c^*_{m,n} h_{\hat{\mathcal{D}}_m}) < \mathcal{L}_{\mathcal{D}_n}(h_{\hat{\mathcal{D}}_n})$.*

*Proof.* Let $c^*_{n,n} = 1$ and $c^*_{m,n} = 0, m \neq n$, then $\sum_{m=1}^N c^*_{m,n} h_{\hat{\mathcal{D}}_m} = h_{\hat{\mathcal{D}}_n}$. In this case,

$$\mathcal{L}_{\mathcal{D}_n}(\sum_{m=1}^N c^*_{m,n} h_{\hat{\mathcal{D}}_m}) = \mathcal{L}_{\mathcal{D}_n}(h_{\hat{\mathcal{D}}_n}). \tag{8}$$

Thus, there always exist $c^*_{m,n}, m = 1, \dots, N$, such that $\mathcal{L}_{\mathcal{D}_n}(\sum_{m=1}^N c^*_{m,n} h_{\hat{\mathcal{D}}_m}) \leq \mathcal{L}_{\mathcal{D}_n}(h_{\hat{\mathcal{D}}_n})$.

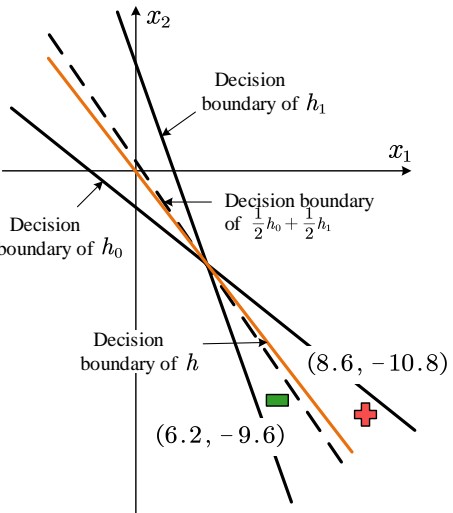

Figure 6: Example of a binary classification task with three different hypothesises. The dashed line denotes the personalized ensemble hypothesis (i.e., $c_{0,0} = c_{0,1} = \frac{1}{2}$). Compared with $h_1$, personalized ensemble model can classify $x_1$ correctly.

Next, we demonstrate that there exist some problems where the personalized ensemble model is strictly better than the local model trained on private data. We consider a binary classification problem with the true decision boundary as $h(x) = \frac{1}{1 + \exp -(4x_1 + 3x_2)}$. Now, consider two clients $\mathcal{C} = \{0, 1\}$ with the related hypothesis as $h_0 = \frac{1}{1 + \exp -(x_1 + x_2 + 1)}, h_1 = \frac{1}{1 + \exp -(x_1 + \frac{1}{2}x_2 - 1)}$ trained on their own private dataset, respectively. It is worthwhile noting that the differences between the client hypothesis and the true hypothesis may be inevitable due to the insufficiency of local dataset. As shown in the Figure 6, the hypothesis of client 0 greatly deviates from the true hypothesis, hence yielding the low classification accuracy. Consider a simple example where there are only two data points in the test dataset, i.e., $x_0 = (6.2, -9.6)$ with label 0 and $x_1 = (8.6, -10.8)$ with label 1. The classification probabilities of client 0 for the two data samples are $P_0(y = 0|x_0) \approx 0.92$ and $P_0(y = 1|x_1) \approx 0.23$,

which indicates that the test accuracy of client $0$ is $50\%$. Now consider the personalized ensemble model of $h_0$ and $h_1$ with $c_{0,0} = c_{0,1} = \frac{1}{2}$. The output probability for two test data samples are $\frac{1}{2}[P_0(y=0|x_0) + P_1(y=0|x_0)] = 0.66$, and $\frac{1}{2}(P_0(y=1|x_1) + P_1(y=1|x_1)) = 0.565$, respectively. The ensemble model has a classification accuracy of $100\%$. Therefore, the personalized ensemble model has better performance than that of the model in client $0$ (i.e., Figure 6), which completes the proof. $\qquad\square$

**Remark A.1.** *There always exist $c_{m,n}^*, m = 1, \ldots, N$, such that the expected loss of the personalized ensemble model for the data distribution $\mathcal{D}_n$ of client $n$ is not larger than that of the average ensemble model: $\mathcal{L}_{\mathcal{D}_n}(\sum_{m=1}^{N} c_{m,n}^* h_{\hat{\mathcal{D}}_m}) \leq \mathcal{L}_{\mathcal{D}_n}(\frac{1}{N}\sum_{m=1}^{N} h_{\hat{\mathcal{D}}_m})$. Besides, there exist some problems where the personalized ensemble model is strictly better, i.e., $\mathcal{L}_{\mathcal{D}_n}(\sum_{m=1}^{N} c_{m,n}^* h_{\hat{\mathcal{D}}_m}) < \mathcal{L}_{\mathcal{D}_n}(\frac{1}{N}\sum_{m=1}^{N} h_{\hat{\mathcal{D}}_m})$.*

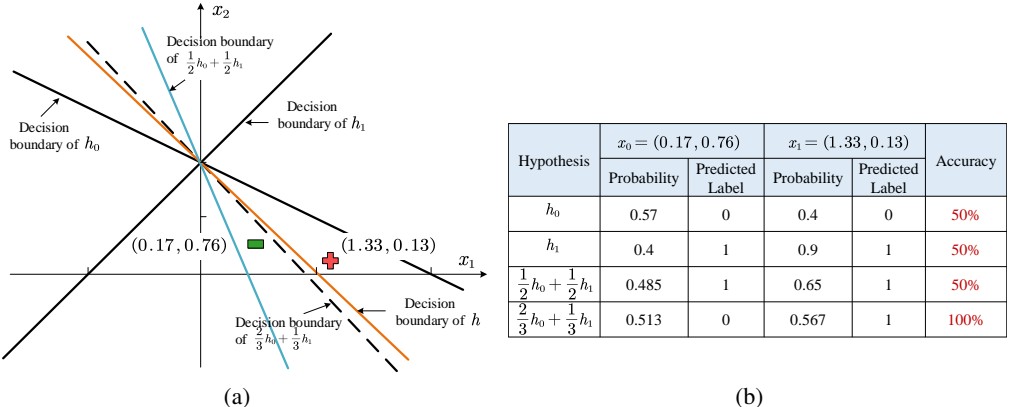

| | | | | | | |
|---|---|---|---|---|---|---|
| Hypothesis | $x_0 = (0.17, 0.76)$ | | $x_1 = (1.33, 0.13)$ | | Accuracy |
| | Probability | Predicted Label | Probability | Predicted Label | |
| $h_0$ | 0.57 | 0 | 0.4 | 0 | 50% |
| $h_1$ | 0.4 | 1 | 0.9 | 1 | 50% |
| $\frac{1}{2}h_0 + \frac{1}{2}h_1$ | 0.485 | 1 | 0.65 | 1 | 50% |
| $\frac{2}{3}h_0 + \frac{1}{3}h_1$ | 0.513 | 0 | 0.567 | 1 | 100% |

(a)                                                 (b)

Figure 7: Example of a binary classification task with three different hypothesises. The dashed line denotes the personalized ensemble hypothesis (i.e., $c_{0,1} = \frac{2}{3}, c_{1,1} = \frac{1}{3}$). Compared with $\frac{1}{2}h_0 + \frac{1}{2}h_1$, personalized ensemble model can classify $x_1$ correctly.

*Proof.* Let $c_{m,n} = \frac{1}{N}, m = 1, \cdots, N$, then $\sum_{m=1}^{N} c_{m,n}^* h_{\hat{\mathcal{D}}_m} = \frac{1}{N}\sum_{m=1}^{N} h_{\hat{\mathcal{D}}_m}$, which demonstrates that there always exist $c_{m,n}^*, m = 1, \ldots, N$, such that $\mathcal{L}_{\mathcal{D}_n}(\sum_{m=1}^{N} c_{m,n}^* h_{\hat{\mathcal{D}}_m}) \leq \mathcal{L}_{\mathcal{D}_n}(\frac{1}{N}\sum_{m=1}^{N} h_{\hat{\mathcal{D}}_m})$.

Similarity, we demonstrate that there exist some problems where the personalized ensemble model is strictly better than the average ensemble model trained on private data. We consider a binary classification problem with the true decision boundary as $h(x) = \frac{1}{1+\exp -(2x_1 - x_2 + 1)}$. Now, consider two clients $\mathcal{C} = \{0, 1\}$ with the related hypothesis as $h_0 = \frac{1}{1+\exp -(x_1 + 2x_2 - 2)}, h_1 = \frac{1}{1+\exp -(x_1 - x_2 + 1)}$ trained on their own private dataset, respectively. As shown in the Figure 7(a), the hypothesis of client $0$ greatly deviates from the true hypothesis, hence yielding the low classification accuracy. Consider a simple example where there are only two data points in the test dataset, i.e., $x_0 = (0.17, 0.76)$ with label 0 and $x_1 = (1.33, 0.13)$ with label 1. The classification probabilities of client $0$ for the two data samples are $P_0(y=1|x_0) \approx 0.43$ and $P_0(y=1|x_1) \approx 0.4$, which indicates that the test accuracy of client $0$ is $50\%$. Now consider the average ensemble model of $h_0$ and $h_1$ with $c_{0,0} = c_{0,1} = \frac{1}{2}$. The output probability for two test data samples are $\frac{1}{2}[P_0(y=0|x_0) + P_1(y=0|x_0)] = 0.485$, and $\frac{1}{2}[P_0(y=1|x_1) + P_1(y=1|x_1)] = 0.65$, respectively. The average ensemble model has a classification accuracy of $50\%$. Further, consider the personalized ensemble model of $h_0$ and $h_1$ with $c_{0,0} = \frac{2}{3}, c_{0,1} = \frac{1}{3}$. The output probability for two test data samples are $\frac{2}{3}P_0(y=0|x_0) + \frac{1}{3}P_1(y=0|x_0) = 0.513$, and $\frac{2}{3}P_0(y=1|x_1) + \frac{1}{3}P_1(y=1|x_1) = 0.567$, respectively. The ensemble model has a classification accuracy of $100\%$. Therefore, the personalized ensemble model have better performance than that of the average ensemble model (i.e., Figure 7(b)), which completes the proof. $\qquad\square$

# B  Additional Experimental Results

In this section, we provide more concrete experimental settings and more numerical results to examine how the hyperparameters affect the performance of our proposed KT-pFL, i.e., number of epochs for local training, distillation steps, regularized parameters and the batch size of public data.

## B.1  Non-IID Data Setting

**Non-IID case 1:** each client contains all classes of samples. We first partition clients into 2 groups, and then assign data samples to clients so that the clients in the same groups have same data distributions, the clients in the different groups have different data distributions. For EMNIST, Fashion_MNIST and CIFAR-10, the number of data samples per class per client in 2 groups is set as [450, 450, 450, 450, 450, 150, 150, 150, 150, 150] and [150, 150, 150, 150, 150, 450, 450, 450, 450, 450], respectively.

**Non-IID case 2:** each client only contains two classes. For all datasets, each client randomly chooses two classes of samples from the whole dataset. The number of samples per class is 300.

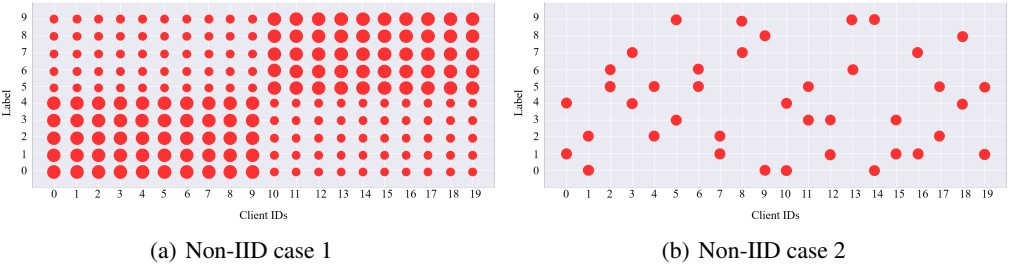

(a) Non-IID case 1  (b) Non-IID case 2

Figure 8: Illustration of the Non-IID data setting.

## B.2  Detailed Experimental Setup

Unless mentioned, otherwise the temperature hyperparameter $T$ is set to 10. The hyperparameter $\lambda$ in Eq (2) and Eq (7) is set as 1.

**Heterogeneous FL.** For all the methods and all the data settings, the batch size on private data and public data are 128 and 256, respectively, the number of local epochs is 20 and the distillation steps is 1 in each communication round of pFL training. Unless mentioned, otherwise the size of public data used in each communication round is 3000, the learning rate (i.e., $\eta_1, \eta_2, \eta_3$) is set to 0.01 for EMNIST and Fashion_MNIST, and 0.02 for CIFAR-10, the regularization parameter is set as 0.6 for EMNIST and Fashion_MNIST, and 0.7 for CIFAR-10.

**Homogeneous FL.** For all the methods and all the data settings, the batch size on private data and public data are 32 and 64, respectively. Unless mentioned, otherwise the number of local epochs is 20 and the distillation steps is 1 in each communication round of pFL training, the learning rate (i.e., $\eta_1, \eta_2$) is set to 0.01 for all datastes, the learning rate for $\eta_3$ is set as 0.005 for EMNIST and Fashion_MNIST, and 0.01 for CIFAR-10.

## B.3  Additional Results

To understand how different hyperparameters such as $E$, $R$, $\rho$, and $|\xi_r|$ affect the training performance of KT-pFL in different settings, we conduct various experiments on three dataset with $\eta_1 = \eta_2 = \eta_3 = 0.01$. In Figure 9, 10, 11 and 12, we visualize the learning curves of KT-PFL under different settings of $E$, $R$, $\rho$, and $|\mathbb{D}_r|$, respectively.

# C  Homogeneous-based KT-pFL Algorithm

To conduct fair comparison with baselines, we exchange model parameters with the server to perform knowledge transfer. Specifically, each client maintains a personalized global model at the server side

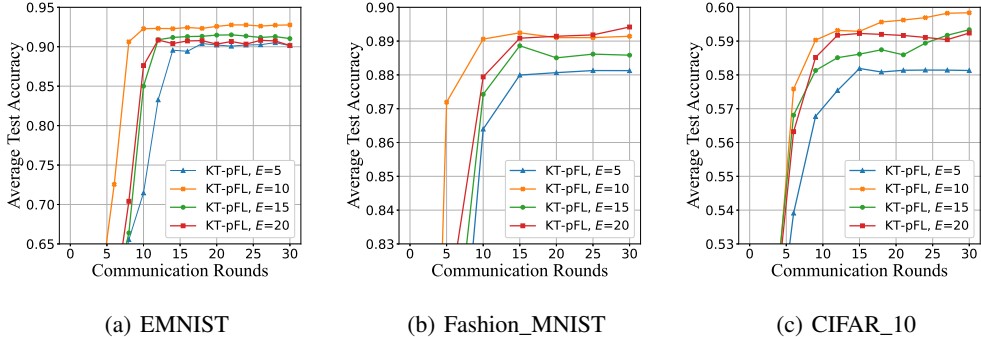

(a) EMNIST      (b) Fashion_MNIST      (c) CIFAR_10

Figure 9: The effect of local epochs on average test accuracy on three datasets.

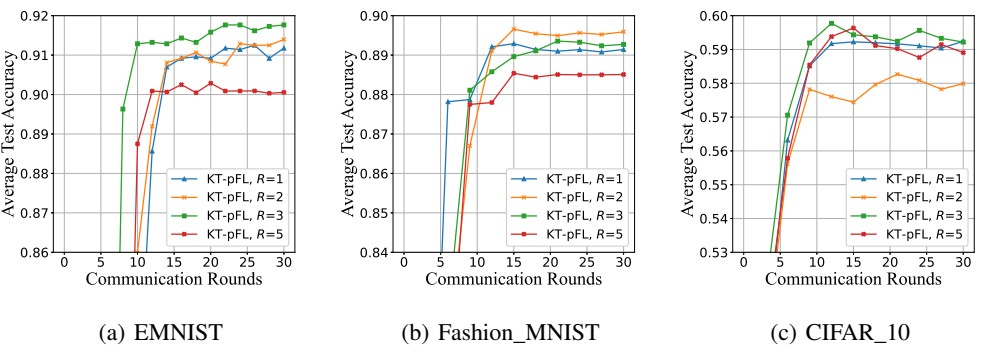

(a) EMNIST      (b) Fashion_MNIST      (c) CIFAR_10

Figure 10: The effect of distillation steps on average test accuracy on three datasets.

and each personalized global model is aggregated by a linear combination of local model parameters and knowledge coefficient matrix. In each round, we fix either $\mathbf{w}$ or $\mathbf{c}$ by turns, and optimize the unfixed one following an alternating way until a convergence point is reached.

in each round we fix either $\mathbf{w}$ or $\mathbf{c}$ by turns, and optimize the unfixed one following an alternating way until a convergence point is reached.

**Update $\mathbf{w}$:** In each communication round, we first fix $\mathbf{c}$ and optimize (train) $\mathbf{w}$ for several epochs locally. In this case, updating $\mathbf{w}$ depends on the private data (i.e., $\mathcal{L}_{CE}$ on $\mathbb{D}_n, n \in [1, \cdots, N]$), that can only be accessed by the corresponding client. We propose a two-stage updating framework for $\mathbf{w}$:

- *Local Training*: Train $\mathbf{w}$ on each client's private data by applying a gradient descent step:
$$\mathbf{w}^n \leftarrow \mathbf{w}^n - \eta_1 \nabla_{\mathbf{w}^n} \mathcal{L}_n(\mathbf{w}^n; \xi_n), \tag{9}$$
where $\xi_n$ denotes the mini-batch of data $\mathbb{D}_n$ used in local training, $\eta_1$ is the learning rate.

- *Parameterized Knowledge Transfer*: After uploading local model parameters, the personalized model for each client is calculated based on current $\mathbf{c}$, then transfer the knowledge from personalized model to each local client:
$$\mathbf{w}^n \leftarrow \mathbf{w}^n - \eta_2 \nabla_{\mathbf{w}^n} \mathcal{L}_{dis} \left( \sum_{m=1}^{N} \mathbf{c}_m^{*,T} \cdot \mathbf{w}^m, \mathbf{w}^n \right), \tag{10}$$
where $\mathcal{L}_{dis}$ denotes the distance function, e.g., MSELoss, and $\eta_2$ is the learning rate. $\mathbf{c}_m^* = [c_{m1}, c_{m2}, \cdots, c_{mN}]$ is the *knowledge coefficient* vector for client $m$, which can be found in $m$-th row of $\mathbf{c}$.

**Update $\mathbf{c}$:** After updating $\mathbf{w}$ locally for several epochs, we turn to fix $\mathbf{w}$ and update $\mathbf{c}$ in the server.
$$\mathbf{c} \leftarrow \mathbf{c} - \eta_3 \lambda \sum_{n=1}^{N} \frac{D_n}{D} \nabla_{\mathbf{c}} \mathcal{L}_{dis} \left( \sum_{m=1}^{N} \mathbf{c}_m \cdot \mathbf{w}^{m,*}, \mathbf{w}^{n,*} \right) - 2\eta_3 \rho (\mathbf{c} - \frac{\mathbf{1}}{N}), \tag{11}$$
where $\eta_3$ is the learning rate for updating $\mathbf{c}$.

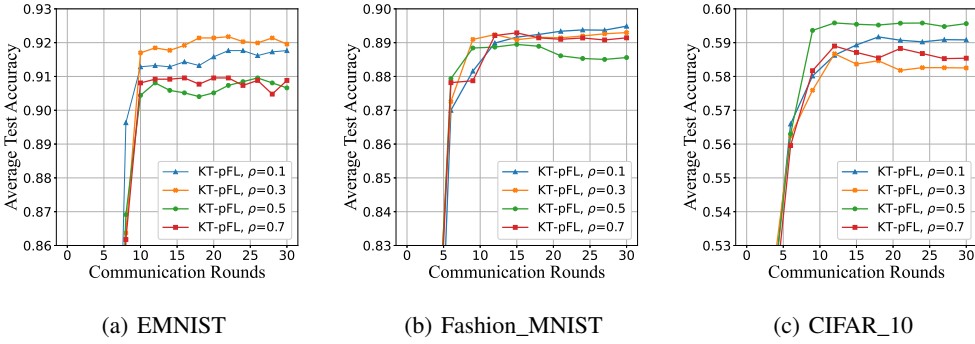

(a) EMNIST  (b) Fashion_MNIST  (c) CIFAR_10

Figure 11: The effect of hyperparameter $\rho$ on average test accuracy on three datasets.

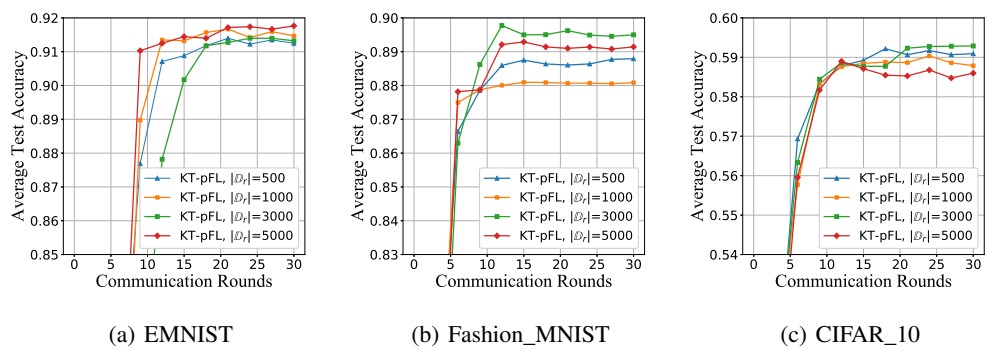

(a) EMNIST  (b) Fashion_MNIST  (c) CIFAR_10

Figure 12: The effect of hyperparameter $|\mathbb{D}_r|$ on average test accuracy on three datasets.

---

**Algorithm 2** Homogeneous-based KT-pFL Algorithm

---

**Input:** $\mathbb{D}$, $\eta_1, \eta_2, \eta_3$ and $T$
**Output:** $\mathbf{w} = [\mathbf{w}^1, \cdots, \mathbf{w}^N]$

1: Initialize $\mathbf{w}_0$ and $\mathbf{c}_0$
2: **procedure** SERVER-SIDE OPTIMIZATION
3:      Distribute $\mathbf{w}_0$ and $\mathbf{c}_0$ to each client
4:      **for** each communication round $t \in \{1, 2, ..., T\}$ **do**
5:          **for** each client $n$ **in parallel do**
6:              $\mathbf{w}_{t+1}^n \leftarrow ClientLocalUpdate(n, \mathbf{w}_t^n, \mathbf{c}_{t,n})$
7:          Update knowledge coefficient matrix $\mathbf{c}$ via (11)
8:          Distribute $\mathbf{c}_{t+1}$ to all clients
9: **procedure** CLIENTLOCALUPDATE$(n, \mathbf{w}_t^n, \mathbf{c}_{t,n})$
10:      Client $n$ receives $\mathbf{w}_t^n$ and $\mathbf{c}_n$ from the server
11:      **for** each local epoch $i$ from 1 to $E$ **do**
12:          **for** mini-batch $\xi_t \subseteq \mathbb{D}_n$ **do**
13:              **Local Training:** update model parameters on private data via (9)
14:      **for** each distillation step $j$ from 1 to $R$ **do**
15:          **Parameterized Knowledge Transfer:** update model parameters via (10)
         **return** local parameters $\mathbf{w}_{t+1}^n$

---