# OpenReview forum: "Parameterized Knowledge Transfer for Personalized Federated Learning"
_NeurIPS.cc/2021/Conference — NeurIPS 2021 Poster_

### Official Review · Reviewer_eENH · 2021-06-28

**Rating:** 4
**Confidence:** 4

**Summary:**

This paper proposes a personalized group knowledge transfer training algorithm (KT-pFL) based on knowledge distillation that deals with model architecture heterogeneity in FL systems.

The authors claim the following contributions:
- This paper is the first to study personalized knowledge transfer in FL.
- Proposing the ‘knowledge coefficient matrix’ to identify the contribution from one client to others’ local training.
- Providing theoretical performance guarantee for KT-pFL and conduct extensive experiments.


**Limitations And Societal Impact:**

The authors should discuss the limitations of their approach in addition to potential negative societal impact.

**Main Review:**

**Pros/Cons**:

[+] The problem this paper aims to solve is novel,  to the best of my knowledge this is the first work to address model architecture heterogeneity in FL.

[+] The paper is well written and structured.

[+] Good theoretical analysis on KT-pFL's performance.

[-] The usage of knowledge coefficient matrix $\textbf{c}$ makes this method unfeasible for large-scale FL systems. $\textbf{c} \in \mathbb{R}^{N \times N}$ where $N$ denotes the number of clients. Previous works in P/FL the Hub/Server transmits the model parameters which can be at extreme case billions of parameters. If we take a small federation size of $10K$ clients we will transmit $0.1$ billion parameters in each COM. The proposed approach will be effective (in terms of communication costs) only on small-scale FL systems.

[-] A major limitation is the usage of public data and its size. In the experiments, the authors used $3000$ samples for their public data which for CIFAR-10 is $5%$ of the data. In Healthcare FL systems it would be hard to obtain such public data, especially in this size. The authors did not provide any discussion related to the public data and did not explain how it was partitioned. Did the baseline methods use these $3000$ samples or they are held out?
I also think that an important addition for this paper should be an experiment that investigates the effect of the public data size on performance.

[-] The authors did not address more recent approaches in PFL [1,2] and did not compare to them.

[-] The update rule of KT-pFL involves three update loops: (i) local training, (ii) distillation, (iii) c update.  This may lead to asynchronous between clients, in FL we aim to mitigate it as it costs time and resources. In addition, it will be interesting to check what is the clock time of KT-pFL compare to FedAvg.

[-] All the clients in the federation take part in each communication round - this is a big limitation for large-scale federations, in real life FL systems federations can contain millions of clients such that training this federation will cost lots of computational power and money. Other PFL methods sample a subset of clients at each COM round (most of them sample 5 clients per round). This limitation raises another question, have the authors run all methods on the same number of communications? each COM round in KT-pFL performs $N$ communications as $N$ denotes the number of clients, while for pFedMe and [1,2] less amount of communications occur per COM round. In FL we care about the number of communications it takes the model to converge.

[-] "the learning rate is set to $0.01$ for EMNIST and Fashion MNIST and $0.02$ for CIFAR-10." It is not clear if the mentioned learning rate values were fixed after HPs search or not and if these values were used for the baselines as well. From the text "All datasets are split randomly with 75% and 25% for training and testing, respectively" I understand that no HPs search was done and these values were used for the baselines as well.

[-] Intuitive baseline for the Hetro FL experiment would be to assign each group of clients (that have different model architecture) with a server and train them separately.

[-] Experiments were conducted on a limited number of clients (20), in real-life FL systems, we have thousands and millions of clients I encourage the authors to run experiments on larger federation sizes e.g. 500 clients.

[-] Missing information on homogenous FL experiment - it is not what "homogenous" refers to if it is only for the model architecture or to the data distribution as well.

[-] The authors should provide STD analysis (i.e. running the experiments on multiple seeds).

**References**:

[1] Shamsian, Aviv, Aviv Navon, Ethan Fetaya, and Gal Chechik. "Personalized Federated Learning using Hypernetworks." arXiv preprint arXiv:2103.04628 (2021).

[2] Zhang, Michael, Karan Sapra, Sanja Fidler, Serena Yeung, and Jose M. Alvarez. "Personalized Federated Learning with First Order Model Optimization." arXiv preprint arXiv:2012.08565 (2020).

**Time Spent Reviewing:**

at least 5 hours

---

> ### Author Response · Authors · 2021-08-10
> **Response to Reviewer eENH (Part 2)**
>
> **Q6**: "the learning rate is set to 0.01 for EMNIST and Fashion MNIST and 0.02 for CIFAR-10." It is not clear if the mentioned learning rate values were fixed after HPs search or not and if these values were used for the baselines as well. From the text "All datasets are split randomly with 75\% and 25\% for training and testing, respectively" I understand that no HPs search was done and these values were used for the baselines as well.
>
> **A6**: It is true that there was no hyper-parameter search in our experiments, the setting of the learning rate and other hyper-parameters (e.g., batch size) is just referred to some existing works, such as FedAMP [8], pFedMe [6].
>
> **Q7**: Intuitive baseline for the Hetro FL experiment would be to assign each group of clients (that have different model architecture) with a server and train them separately.
>
> **A7**: As suggested by the reviewer, we have added new baselines — original FedDF [16], whose idea is to aggregate the clients with the same architecture to a group, and form a global model in this group, KD is also used to further improve the performance of the global models. The detailed experimental results can be seen below:
>
> | Method | EMNIST |  | FashionMNIST| | CIFAR-10 | |
> | :-----| :----: | :----: |  :----: |  :----: |  :----:  |  :----: |
> | | Non-IID_1 | Non-IID_2 | Non-IID_1| Non-IID_2|  Non-IID_1| Non-IID_2|
> | fedMD [13] | 72.75 | 74.34 | 71.89| 72.76| 42.03| 39.50|
> | FedDF [16] | 72.89|75.41 |72.32| 72.83| 42.48| 39.55|
> | pFedDF | 78.59| 75.72| 72.69| 73.45| 42.83| 39.54|
> |Sim-pFD| 82.72|74.42|72.82|72.71|42.50|39.90|
> |TopK-pFD|83.00| 73.91|72.24|73.39|42.58|39.34|
> |KT-pFL|84.65|76.30|75.48|75.60|43.82|41.04|
>
> [16] Tao Lin, Lingjing Kong, Sebastian U. Stich, and Martin Jaggi. Ensemble distillation for
> robust model fusion in federated learning. NeurIPS, 2020.
>
>
> **Q8**: Experiments were conducted on a limited number of clients (20), in real-life FL systems, we have thousands and millions of clients I encourage the authors to run experiments on larger federation sizes e.g. 500 clients.
>
> **A8**: Thanks for your detailed suggestion, new experiments with 100 clients have been added, the detailed response can be referred to **A5**.
>
> **Q9**: Missing information on homogeneous FL experiment - it is not what "homogeneous" refers to if it is only for the model architecture or to the data distribution as well.
>
> **A9**: We agree that the actual meaning of “homogeneous” refers to many aspects, which include but are not limited to model architecture, data distribution, and even computational/communication resources. We are sorry for the informal “homogeneous” statement in the paper. We should mention that the meaning of “homogeneous FL” in Line 248 is the FL system with a homogeneous model architecture.
>
> **Q10**: The authors should provide STD analysis (i.e. running the experiments on multiple seeds).
>
> **A10**: If we understand correctly, “STD analysis” means standard deviation analysis. In our experiments, we run each experiment multiple times and record the average results.

---

> ### Author Response · Authors · 2021-08-10
> **Response to Reviewer eENH (Part 1)**
>
> We would like to thank the reviewer for the constructive and detailed comments, we have tried to address all your concerns one by one. We would also like to highlight that,
> in our response, we have added some new baselines to further demonstrate the efficiency of our proposed method. Besides that, we have conducted additional experiments in large-scale FL systems, The detailed response for each question can be seen below:
>
> **Q1**: The usage of knowledge coefficient matrix c makes this method unfeasible for large-scale FL systems. $c \in \mathbb{R}^{N \times N}$ where N denotes the number of clients. Previous works in P/FL require the Hub/Server to transmit the model parameters which can be up to billions of parameters. If we take a small federation size of 10K clients we will transmit 0.1 billion parameters in each COM. The proposed approach will be effective (in terms of communication costs) only on small-scale FL systems.
>
> **A1**: Thanks for the review but there seems to be a misunderstanding about the knowledge coefficient matrix $\mathbf{c}$.
> We want to point out that the
> knowledge coefficient matrix $\mathbf{c}$ will not be transmitted between clients and server. In fact, the clients in KT-pFL only need to upload the local soft-predictions (i.e., the size is related to the number of classes for a specific task) and download the personalized soft-prediction to/from the server in each communication round. The knowledge coefficient matrix is stored and updated in the server, with no need to interact with the clients. The public dataset can be downloaded from the server to each client just before the training starts. Therefore, our proposed method avoids the substantial communication overhead caused by model parameters transmission in traditional pFL works. The detailed comparison on communication overhead can be seen in Figure 5. Moreover, we clarify that KT-pFL can also be effective on large-scale FL systems since the size of the soft predictions is far less than that of model parameters. In other words, client sampling can also be used in KT-pFL on large-scale FL systems. In this case, partial clients will be selected and related partial of elements in knowledge coefficient matrix $\textbf{c}$ will be updated in each communication round.
>
> **Q2**: A major limitation is the usage of public data and its size. In the experiments, the authors used 3000 samples for their public data which for CIFAR-10 is 5 of the data. In Healthcare FL systems, it would be hard to obtain such public data, especially in this size. The authors did not provide any discussion related to the public data and did not explain how it was partitioned. Did the baseline methods use these 3000 samples, or they are held out? I also think that an important addition for this paper should be an experiment that investigates the effect of the public data size on performance.
>
> **A2**: We agree with the reviewer that in some cases the labeled public dataset is hard to obtain. However, in these cases, we can use some unlabeled open datasets and even synthesized data generated by a GAN before the training starts, which are easy to obtain.
> Another advantage of using unlabeled open datasets and synthesized public data is the ability to avoid potential privacy leakage of the private local data.
> Therefore, the public dataset is not a big bottleneck in our proposed method,
> as there are already many existing works demonstrate that the diversity of the public data does not significantly impact the performance of distillation [i.e., 16].
> We are sorry for the insufficient explanation on public data construction and partition scheme. In our proposed training methods, the public dataset can be stored in the server, so each client will download the same set of public data samples (i.e., 3000 in our experiments) before training. To be fair, each KD-based baseline method uses the same set of public data samples. We have already investigated the effect of the public dataset size on performance, which can be seen on the right side of Table 4 (we are sorry we use a wrong statement $|\xi_r|$, which should be corrected to $|\mathbb{D}_r|$).
>
> **Q3**: The authors did not address more recent approaches in PFL [1,2] and did not compare to them.
>
> **A3**: Thanks for your valuable comments. We have added some context to discuss these pFL works[1,2]. Specifically, similar to FedAMP [8], pFedHN (i.e., [1]) and FedFomo (i.e., [2]) still exchange model parameters among the server and clients. Beyond that, FedFomo maintains a personalized model for each client in the server in a non-parametrized manner, where the combination weight of each local model is calculated by a similarity function, which requires each client to download other clients’ model locally, additional communication overhead and potential privacy leakage cannot be neglected. As for pFedHN [1], an embedding vector for each client is required for the hypernetwork in the server, but these embedding vectors that show the client representation would cause privacy issues to some extent.
> As suggested by the reviewer, we have added **pFedHN** [1] and **FedFomo** [2] as two new baselines for further comparison. The detailed experiments can be seen below:
>
> | Method | EMNIST| FashionMNIST| CIFAR-10 |
> | :-----| :----: |  :----: |  :----: |
> |Fedavg | 85.86| 80.69| 51.94|
> |Per-Fedavg | 86.16| 85.91| 55.61|
> |pFedMe | 84.26| 90.97| 62.79|
> |FedAMP [8]| 88.51| 90.79| 60.75|
> | FedFomo [2]| 92.58| 88.33| 63.11|
> |pFedHN [1]| 94.27| 92.22| 65.31|
> |KT-pFL| 94.40| 93.93| 66.96|
>
> [1] Shamsian, Aviv, Aviv Navon, Ethan Fetaya, and Gal Chechik. "Personalized Federated Learning using Hypernetworks." arXiv preprint arXiv:2103.04628 (2021).
>
> [2] Zhang, Michael, Karan Sapra, Sanja Fidler, Serena Yeung, and Jose M. Alvarez. "Personalized Federated Learning with First Order Model Optimization." arXiv preprint arXiv:2012.08565 (2020).
>
> **Q4**: The update rule of KT-pFL involves three update loops: (i) local training, (ii) distillation, (iii) $\textbf{c}$ update. This may lead to asynchronous between clients, in FL we aim to mitigate it as it costs time and resources. In addition, it will be interesting to check what is the clock time of KT-pFL compare to FedAvg.
>
> **A4**: We agree with the reviewer that asynchronization between clients will arise in our proposed training framework in some cases. The reasons may include but are not limited to the heterogeneous computational/communication speed, non-IID data, etc.
> However, we apply a synchronous strategy in our proposed KT-pFL algorithm, which guarantees the local training and distillation phase in each client is synchronous and uses the same knowledge coefficient matrix $\textbf{c}$ in each communication round.
> We can still try to consider the clock time of KT-pFL.
> When compared with FedAvg, the additional computation can be summarized into two aspects: the distillation phase in client-side; calculation and update of the knowledge coefficient matrix $\mathbf{c}$ on the server-side. If we ignore the time spend on the high-performance server, the time in the distillation phase will rely heavily on the computational resource and the public data size of each client.
>
> **Q5**: All the clients in the federation take part in each communication round - this is a big limitation for large-scale federations, in real life FL systems federations can contain millions of clients such that training this federation will cost lots of computational power and money. Other PFL methods sample a subset of clients at each COM round (most of them sample 5 clients per round). This limitation raises another question, have the authors run all methods on the same number of communications? each COM round in KT-pFL performs $N$ communications as $N$ denotes the number of clients, while for pFedMe and [1,2] less amount of communications occur per COM round. In FL we care about the number of communications it takes the model to converge.
>
> **A5**: Sorry for insufficient experiments on large-scale FL systems. We have added new experiments on 100 clients and applied the same client selection mechanism as other baselines. We clarify that our proposed method can work well both on small-scale and large-scale FL systems, the key difference between these two scenarios is that only partial of elements in the knowledge coefficient matrix $\textbf{c}$ will be updated in each communication round on large-scale FL systems. Furthermore, we have added new experiments to compare the required communication rounds to achieve convergence for each method. The detailed results can be seen below:
>
> |Method| Fedavg| Per-fedavg| pFedMe| FedAMP| FedFomo| pFedHN| KT-pFL|
> | :-----| :----: |  :----: |  :----: |  :----: |  :----: |   :----: |  :----: |
> |EMNIST|76.84|82.87|86.14|85.63|90.08|92.21|92.50|
> |FashionMNIST|72.36|77.93|84.02|84.72|87.76|89.14|90.37|
> |CIFAR-10| 43.56| 49.82| 52.46| 51.26| 55.73| 57.38| 58.29|

---

> ### Author Response · Authors · 2021-09-01
> **Additional Response to Reviewer eENH**
>
> We hope that our response can address your concerns properly and we will appreciate it if you have more comments and suggestions.

---

### Official Review · Reviewer_PX5C · 2021-07-16

**Rating:** 6
**Confidence:** 4

**Summary:**

The paper studies personalized federated learning (pFL), where the local models have different structures and sizes. The proposed method enables each client to maintain a personalized soft prediction at the server-side. Empirical results show the strength of the proposed solution.

**Limitations And Societal Impact:**

Yes

**Main Review:**

The paper proposes a knowledge coefficient matrix at the server side to guide the local training. The pFL is a timely topic and the proposed approach is also interesting.

Originality: This work is a novel combination of well-known techniques.
Quality: The submission is technically sound, and all claims are well supported.
Clarity: The submission is clearly written and well organized.
Significance: The others (researchers or practitioners) are likely to use the ideas or build on them.

Strengths:
1.	Writing is clear. Details of each component are explained and reproducible from the appendix and the code.
2.	The methodology is reasonable. The proposed knowledge coefficient matrix is well-designed with intuitive justifications.
3.	This paper presents non-trivial performance improvement against SOTA pFL methods on 3 popular benchmark datasets.

Weaknesses:
1.	The introduction of the knowledge coefficient matrix may raise the concern of privacy leaks and the communication cost.
2.	The use of the public data in step 2 in Figure 1 conflicts with the general FL setting.
3.	To my best knowledge, there are some other personalized FL works (Hierarchical Personalized Federated Learning for User Modeling, WWW'21; Exploiting Shared Representations for Personalized Federated, arXiv:2102.07078), which need to be listed as baselines.


**Time Spent Reviewing:**

5 hours

---

> ### Author Response · Authors · 2021-08-10
> **Response to Reviewer PX5C**
>
> We thank you for the constructive comments. We have tried to address all your comments and suggestions on **Weaknesses** as follows:
>
> **Q1**: The introduction of the knowledge coefficient matrix may raise the concern of privacy leaks and the communication cost.
>
> **A1**: **Knowledge coefficient matrix.** In our proposed method, the knowledge coefficient matrix is stored and updated on the server-side, none of the clients need to download the knowledge coefficient matrix during the whole training process. As shown in Figure 1, the clients just need to upload local soft-predictions and download the personalized soft predictions to/from the server. In this case, there is no additional communication cost in the whole workflow, and privacy can be guaranteed.
>
> **Q2**: The use of the public data in step 2 in Figure 1 conflicts with the general FL setting.
>
> **A2**: **Use of the public dataset.**
> Sorry for the insufficient explanation of this part. In practice, the public dataset can be stored in the server, then each client downloads the public dataset to the local machine for distillation. Besides, the public dataset can both be labeled dataset, unlabeled open dataset, and even synthesized data generated by a GAN, which relies little on private data information, and no private data are needed to transmit to the server. We clarify that our proposed training framework works well in both cases and is perfectly compatible with the FL setting.
>
> **Q3**: To my best knowledge, there are some other personalized FL works (Hierarchical Personalized Federated Learning for User Modeling, WWW'21; Exploiting Shared Representations for Personalized Federated, arXiv:2102.07078), which need to be listed as baselines.
>
> **A3**: **More baselines.**  As suggested by the reviewer, we have added some new baselines in heterogeneous systems and homogeneous systems, respectively. More specifically, we have added **FedMD** [13] and **pFedDF** (i.e., FedDF+ local fine-tuning, which is designed for pFL) as new baselines on heterogeneous experiments; In terms of homogeneous system, we have added **FedFomo**[1] and **pFedHN** [2] as two new baselines. Besides, we would like to explain that both [3] and [4] are only applicable to limited cases with special neural network architectures, and by such means, we do not add these two works as new baselines in our additional experiments. The detailed experimental results can be seen below:
>
> - In heterogeneous FL:
>
> | Method | EMNIST |  | FashionMNIST| | CIFAR-10 | |
> | :-----| :----: | :----: |  :----: |  :----: |  :----:  |  :----: |
> | | Non-IID_1 | Non-IID_2 | Non-IID_1| Non-IID_2|  Non-IID_1| Non-IID_2|
> | fedMD [13] | 72.75 | 74.34 | 71.89| 72.76| 42.03| 39.50|
> | FedDF [16] | 72.89|75.41 |72.32| 72.83| 42.48| 39.55|
> | pFedDF | 78.59| 75.72| 72.69| 73.45| 42.83| 39.54|
> |Sim-pFD| 82.72|74.42|72.82|72.71|42.50|39.90|
> |TopK-pFD|83.00| 73.91|72.24|73.39|42.58|39.34|
> |KT-pFL|84.65|76.30|75.48|75.60|43.82|41.04|
>
>
> - In homogeneous FL:
>
> | Method | EMNIST| FashionMNIST| CIFAR-10 |
> | :-----| :----: |  :----: |  :----: |
> |Fedavg | 85.86| 80.69| 51.94|
> |Per-Fedavg| 86.16| 85.91| 55.61|
> |pFedMe | 84.26| 90.97| 62.79|
> |FedAMP | 88.51| 90.79| 60.75|
> | FedFomo [1]| 92.58| 88.33| 63.11|
> |pFedHN [2]| 94.27| 92.22| 65.31|
> |KT-pFL| 94.40| 93.93| 66.96|
>
> [1] Personalized federated learning with first-order model optimization, ICLR 2021
>
> [2] Personalized Federated Learning using Hypernetworks, ICML 2021
>
> [3] Hierarchical Personalized Federated Learning for User Modeling, WWW'21
>
> [4] Exploiting Shared Representations for Personalized Federated, arXiv:2102.07078
>
> [13] Daliang Li and Junpu Wang. FedMD: Heterogenous federated learning via model distillation. arXiv, oct 2019.
>
> [16] Tao Lin, Lingjing Kong, Sebastian U. Stich, and Martin Jaggi. Ensemble distillation for
> robust model fusion in federated learning. NeurIPS, 2020.
>
> We hope that our response can address your concerns properly and we will appreciate it if you have more suggestions.

---

> > ### Comment · Reviewer_PX5C · 2021-08-31
> > **Rebuttal Feedback**
> >
> > Thanks for your response and for adding the experiment results and related work (Additional Response to Reviewer 21U2 (Related Work)). I have no more questions.

---

### Official Review · Reviewer_21U2 · 2021-07-18

**Rating:** 6
**Confidence:** 5

**Summary:**

This paper works on personalized federated learning (FL). The authors specifically studied the situation that clients train different model architectures. The authors leveraged additionally, public unlabeled data and knowledge distillation (KD) to aggregate these heterogenous clients' models specifically for each client, so that each client can learn their personalized model with the help from other clients. Experimental results on several datasets show the effectiveness of the proposed method.

**Ethical Concerns:**

No concerns.

**Limitations And Societal Impact:**

The authors didn't discuss the limitation. However, this does not affect my score. I don't think the paper will have potential negative societal impacts.

**Main Review:**

I number my questions or the paper's weaknesses as follows. If possible, I would like the authors to at least address 1-4 and 7-8.

======= *Originality*: The proposed method is somewhat a novel combination of (1) KD for FL and (2) deriving a personalized global model/regularizer for FL. However, the authors did not explicitly state these relationships to existing works clearly. Instead, the authors mentioned some of the relationships distributedly in several places of the paper. Besides, many related works are not cited, discussed, or empirically compared. All these factors largely limit the papers’ originality.

1. There have been a few FL works on heterogeneous systems, e.g., [a, b]. Specifically, [b] has a title similar to the proposed method. Both papers are not cited. Also, there is no description in the related work on existing solutions to heterogeneous systems (e.g., [16]), giving a wrong impression that the proposed method is the first to handle this situation.

[a] HeteroFL: Computation and Communication Efficient Federated Learning for Heterogeneous Clients, ICLR 2021

[b] Group Knowledge Transfer: Federated Learning of Large CNNs at the Edge, NeurIPS 2020

2. Knowledge distillation (KD) is one promising way to handle system heterogeneity. However, instead of discussing how KD has been used in FL in the related work (e.g., [12-16]), the authors just provided a general overview of KD. Specifically, [13] is on personalized FL but without any discussion. Moreover, several other FL works using KD are not cited and discussed [c-f].

[c] FedBE: Making Bayesian Model Ensemble Applicable to Federated Learning, ICLR 2021

[d] Distilled one-shot federated learning, arXiv 2020

[e] Federated Model Distillation with Noise-Free Differential Privacy, arXiv 2020

[f] One-Shot Federated Learning, arXiv 2019

3. Another important component is personalized knowledge transfer. It seems that [8, 24] share similar ideas to the proposed method, but the authors didn’t provide a detailed discussion. Some other similar works also derive personalized global models or regularizers [g, h].

[g] Personalized federated learning with first order model optimization, ICLR 2021

[h] Federated Mixture of Experts, OpenReview 2020

Without a clear discussion of these related works and a discussion on the challenge of a naïve combination of them, it is hard for me to assess the paper's originality.

======= *Quality*: The proposed method is technically sound. The optimization procedure is very clean and should be easily reproducible. The authors also conduct extensive experiments to justify the effectiveness of the proposed algorithm. However, there is no clear description and motivation on why it is designed. Also, some important baselines and experimental details are missing; some experimental designs can be improved. See the following.

4. More specifically, the authors suddenly gave a formulation in Eq (2) without giving the design principle. I would suggest that the authors first describe an objective function for personalized FL, and then gradually lead to Eq (2) for the heterogeneous case. Also, I don’t fully understand the description in Line 146-147 for the regularization term in Eq (4).

5. There seems to be no clear description of how the public data is constructed. Does each communication round use the same public data? Also, there is no study on the size of the public data (not merely the batch size).

6. Experimental designs. I checked the design of "each client contains all classes of samples" in the supplementary. It does not lead to a strong non-IID case. Please take a look at [c, i] and consider re-design the experiments

[i] Measuring the effects of non-identical data distribution for federated visual classification, arXiv 2019

7.	Experiments on heterogeneous systems. Can the authors provide more details on FedDF, which was not designed for personalized FL? Specifically, do the authors compare to the distilled model at the server or the local models after local update/fine-tuning? It is very important to compare to the local models. Can the authors compare to the original FedDF (and with local updates before evaluating on personalized data), instead of the modified version in footnote 2? Note that, FedDF does not add any regularized term like the one in Eq (2) in its local training. Also, [13] should be compared. These experiments are very important to ensure a fair and comprehensive comparison. Besides, can the authors further compare to a version of Eq (2) with identical c_m,n? That is, there is no personalized knowledge transfer.

8.	Experiments on homogeneous systems. First, I don’t see a reason why the authors have to modify their algorithm for this setup (Line 249-252). Is this a limitation of the work? Second, the baselines in Table 2 are not very strong. Please consider comparing to some of [i, j, g]. Moreover, please compare to FedAvg after local fine-tuning [17, k]. Third, the proposed method has an obvious advantage --- the public data. Thus, I would suggest that the authors further compare methods based on KD (e.g., [13, 16, c]). For those not designed for personalized FL, the authors can simply fine-tune the global models to get personalized models. Finally, the accuracy of CIFAR-10 in Table 2 is fairly low.  Do the authors know why?

[i] Think locally, act globally: Federated learning with local and global representations, arXiv 2020

[j] Federated learning with personalization layers, arXiv 2019

[k] Salvaging Federated Learning by Local Adaptation, arXiv 2020

======= *Clarity*: The paper is well-written and organized.
Significance: The authors work on an important problem, federated learning under the heterogeneous system setting. I think the proposed method could serve as a strong baseline to be compared or built upon in the future. However, the significance of the paper is limited by the aforementioned questions.

**Post-rebuttal update**
I thank the authors for their tremendous efforts in addressing my concerns for multiple rounds. Many of my concerns are addressed so I increased the score from 4 to 6. I encourage the authors to follow my latest response to incorporate the feedback and their rebuttal into the final version.

**Time Spent Reviewing:**

4 hours

---

> ### Author Response · Authors · 2021-08-10
> **Response to Reviewer 21U2 (Part 2)**
>
> **3. Response to Question 7-8**
>
> Before answering the question 7 and 8, we would like to explain the experimental design and the baseline selection of our paper.
> Since our proposed KT-pFL aims to achieve personalized knowledge distillation in a parameterized way, we conduct extensive experiments both on heterogeneous and homogeneous FL systems to show the efficiency of our proposed training framework. In heterogeneous systems, we compared the performance of KT-pFL with the state-of-the-art KD-based FL training methods (note that all methods exchange soft-predictions with the server instead of model parameters). In homogeneous cases, we compare the performance of KT-pFL with the state-of-the-art pFL training methods, and to be fair, all the baselines exchange model parameters with the server. The experimental results show that KT-pFL works well both on heterogeneous and homogeneous cases.
>
> **Q7**: Experiments on heterogeneous systems. Can the authors provide more details on FedDF, which was not designed for personalized FL? Specifically, do the authors compare to the distilled model at the server or the local models after local update/fine-tuning? It is very important to compare to the local models, Can the authors compare to the original FedDF (and with local updates before evaluating on personalized data), instead of the modified version in footnote 2? Note that, FedDF does not add any regularized term like the one in Eq (2) in its local training. Also, [13] should be compared. These experiments are very important to ensure a fair and comprehensive comparison. Besides, can the authors further compare to a version of Eq (2) with identical $c_{m,n}$? That is, there is no personalized knowledge transfer.
>
> **A7**: Firstly, FedDF [16] is not designed for pFL. Specifically, FedDF requires each client to upload the model parameters to the server for aggregation. Then, the knowledge distillation between the global model and local models is performed in the server.
> As suggested by the reviewer, we have added two new baselines which can solve the concerns listed above: original **FedDF** [16] (corresponding to distilled model at the server), **pFedDF** (FedDF+local fine-tuning, corresponding to the local models after local update/fine-tuning), **FedMD** [13] (corresponding to a version of Eq (2) with identical $c_{m,n}=\frac{1}{N}$).
> From the experimental results, we can further demonstrate the efficiency of our proposed parameterized method in pFL.
> The detailed results can be shown in the following table:
>
> | Method | EMNIST |  | FashionMNIST| | CIFAR-10 | |
> | :-----| :----: | :----: |  :----: |  :----: |  :----:  |  :----: |
> | | Non-IID_1 | Non-IID_2 | Non-IID_1| Non-IID_2|  Non-IID_1| Non-IID_2|
> | fedMD [13] | 72.75 | 74.34 | 71.89| 72.76| 42.03| 39.50|
> | FedDF [16] | 72.89|75.41 |72.32| 72.83| 42.48| 39.55|
> | pFedDF | 78.59| 75.72| 72.69| 73.45| 42.83| 39.54|
> |Sim-pFD| 82.72|74.42|72.82|72.71|42.50|39.90|
> |TopK-pFD|83.00| 73.91|72.24|73.39|42.58|39.34|
> |KT-pFL|84.65|76.30|75.48|75.60|43.82|41.04|
>
>
> [13] Daliang Li and Junpu Wang. FedMD: Heterogenous federated learning via model distillation. arXiv, oct 2019.
>
> [16] Tao Lin, Lingjing Kong, Sebastian U. Stich, and Martin Jaggi. Ensemble distillation for robust model fusion in federated learning, NeurIPS, 2020.
>
> **Q8**: Experiments on homogeneous systems. First, I don’t see a reason why the authors have to modify their algorithm for this setup (Line 249-252). Is this a limitation of the work? Second, the baselines in Table 2 are not very strong. Please consider comparing to some of [i, j, g]. Moreover, please compare to FedAvg after local fine-tuning [17, k]. Third, the proposed method has an obvious advantage --- the public data. Thus, I would suggest that the authors further compare methods based on KD (e.g., [13, 16, c]). For those not designed for personalized FL, the authors can simply fine-tune the global models to get personalized models. Finally, the accuracy of CIFAR-10 in Table 2 is fairly low. Do the authors know why?
>
> [i] Think locally, act globally: Federated learning with local and global representations, arXiv 2020
>
> [j] Federated learning with personalization layers, arXiv 2019
>
> [k] Salvaging Federated Learning by Local Adaptation, arXiv 2020
>
> **A8**:
> In homogeneous FL systems, we focus on comparing the state-of-the-art pFL methods with our proposed method and demonstrating that the concept of parameterized knowledge coefficient matrix can work well in more general cases. Therefore, we modify the original prediction-based algorithm into a parameter-based version to show the generality of our proposed method. Secondly, we agree with the reviewer that [i, j, g], FedAvg after local fine-tuning [17, k] and other KD-based methods [13, 16] should be compared. To sum up, we have added two new baselines: **FedFomo** [g]; **pFedHN** [m]. Note that pFedHN [m] has already demonstrated the advantages over LG-FedAvg [i] and FedPer [j]; [13, 16] have been already compared in heterogeneous FL systems (see Table in **A7**); Per-Fedavg [9] can represent the FedAvg after local fine-tuning; [c] is not designed for pFL so that we ignore this method as a baseline. We can see from the experimental results that our proposed KT-pFL method still has the best performance when compared with other methods. The detailed results can be shown in the following table:
>
> | Method | EMNIST| FashionMNIST| CIFAR-10 |
> | :-----| :----: |  :----: |  :----: |
> |Fedavg [1]| 85.86| 80.69| 51.94|
> |Per-Fedavg [9]| 86.16| 85.91| 55.61|
> |pFedMe [6]| 84.26| 90.97| 62.79|
> |FedAMP [8]| 88.51| 90.79| 60.75|
> | FedFomo [g]| 92.58| 88.33| 63.11|
> |pFedHN [m]| 94.27| 92.22| 65.31|
> |KT-pFL| 94.40| 93.93| 66.96|
>
> Finally, the reason for the low accuracy of CIFAR-10 in original Table 2 is because we only use partial samples instead of total samples to conduct experiments, we are sorry for the confusion and unclear explanation. We have rerun the experiments on CIFAR-10, and illustrated the new results in the above table.
>
> [m] Personalized Federated Learning using Hypernetworks, ICML 2021

---

> > ### Comment · Reviewer_21U2 · 2021-08-22
> > **Post-rebuttal feedback**
> >
> > I thank the authors' tremendous efforts in responding to my comments. The authors have made attempts to answer all my questions. However, there are still several concerns that remained unsolved.
> >
> >
> > For A1-A3, the authors seem to explain that they didn't cite/discuss those related works because they did not work on the exact setting. I respectfully do not agree with the reason. In my opinion, for a paper to be accepted, the authors should cite those papers and provide a clear discussion of why they are not applicable/feasible to the proposed setting --- which can then be seen as the motivation. As I mentioned, without citing and discussing them in the first place, a reader (or even a reviewer) may get the wrong impression that (a) dealing with heterogeneous systems (b) using KD in FL (c) creating personalized global model/regularizer are all new to the FL community. Instead, each of them has been explored earlier.
> >
> >
> > Also for A2, the authors wrote "Strictly speaking, [13] is not designed for personalized FL, but just for handling heterogeneous FL as [13] produces a single global soft prediction for each client." I respectfully do not agree. In Algorithm 1 of [13], their outputs are exactly personalized models f_k for each client. The fact that [13] produces a single public prediction for KD doesn't conflict with the fact that [13] is a personalized algorithm since f_k will be updated locally with local data. Note that, many existing pFL algorithms also create a single regularizer, like [6]. I don't think creating personalized regularization or personalized targets of KD is necessary for a method to be called a personalized FL algorithm.
> >
> >
> > For Q5, I'm indeed asking what public data are used for each experiment. I don't think the authors use the GAN-generated data. Do the authors save some data of the same dataset, like a portion of CIFAR-10, as the public data for the CIFAR-10 experiment? I may miss some details, but can the authors point me to the lines for these details?
> >
> >
> > For Q7, my concern is that [16] can indeed be applied in a heterogeneous system setting --- it does not average the local models, and footnote 2 of the authors' current manuscript seems to be wrong. I understand the "fair comparison" purpose by the authors, but I do want to ask if there is any technical/performance difference between exchanging the predictions or the models for [16]. I appreciate the authors' response in A7, especially with extra experiments. But, some of my questions are not addressed --- specifically, I'd like to know what the exact details are for [16] in Table 1 of the paper. (Please see my original question.) The numbers there are not in the newly provided table. Are the new FedDF and pFedDF in the new table based on exchanging models or predictions? Also, I have doubts about the pFedDF results --- in my experience and several pFL algorithms based on fine-tuning, the results will be significantly better than FedDF. More specifically, can the authors simply take the trained local models for each client at the last round of FedDF as the pFedDF models, instead of taking the final FedDF global model and fine-tuning it with just a few steps?
> >
> >
> > For Q8, I have a similar question as Q7. Is there any technical or performance difference for the proposed method by exchanging the predictions or the models? I just realize that there seems to be no algorithm (like Algorithm 1) for the proposed method with exchanging model parameters. Can the authors clarify the algorithm and what the difference is from Algorithm 1? If exchanging the predictions or models has no fundamental difference for the proposed methods, I think the authors do not need to specifically mention Line 249-252, which indeed causes confusion. But if there is a difference, I'd indeed want to see the results based on exchanging model parameters to make the proposed method more consistent.
> >
> > Besides, I do want to see the results based on fine-tuning. More specifically, can the authors report the results of FedAvg --- but instead of using its global models, using its local models. I respectfully do not agree that "Per-Fedavg [9] can represent the FedAvg after local fine-tuning" since [9] only updates the global model for a few steps, not like the local models of FedAvg (and [17, k]) which are updated with local data for epochs. I also want the authors to compare algorithms with public data here. As mentioned, the reason is that "Third, the proposed method has an obvious advantage --- the public data." Alternatively, the authors can extend the experiments in the right part of Table 4 by considering even fewer public data. I believe at a certain point, the results will degrade. To clarify, it is not a bad thing, but it will be great to let the reader know the benefit of the public data. For example, if there are only 10 public data points, what will be the result?
> >
> >
> > **Minor:**
> >
> > For A4, the authors could first discuss some regularization-based pFL algorithms --- to emphasize that existing methods have indeed included an extra term in pFL beyond the local empirical risk. The authors can then motivate their second term in Eq (2) as a way to regularize the local training.
> >
> >
> > For Q6, it will be definitely to see those more strong diverse data across clients.
> >
> > ===========
> > According to the current rebuttal, I keep my score unchanged.

---

> > > ### Author Response · Authors · 2021-08-28
> > > **Additional Response to Reviewer 21U2  (Related Work)**
> > >
> > > ### **Related Work**
> > >
> > > **Personalized FL with Homogeneous Models.** Recently, various approaches have been proposed to realize personalized FL with homogeneous local model structure, which can be categorized into three types according to the number of global models applied in the server, i.e., single global model, multiple global models and no global model.
> > >
> > > single global model type is a close variation of conventional FL, e.g., FedAvg [1], that combine global model optimization process with additional local model customization, and consist of four different kinds of approaches: local fine-tuning [2-5], regularization (e.g., pFedMe [6], L2SGD [7,8], Ditto [9]),
> > > hybrid local and global models [10-12] and meta learning [13,14]. All of these pFL methods apply a single global model, and thus limit the customized level of the local model at the client side. Therefore, some researchers [15,16,17] propose to train multiple global models at the server, where clients are clustered
> > > into several groups according to their similarity and different models are trained for each group.
> > > FedAMP [15] and FedFomo [17] can be regarded as special cases of the clustered-based method that each client owns a personalized global model at the server side.
> > > As a contrast, some literature waive the global model to deal with the heterogeneity problem [18,19] such as multi-task learning based (i.e., MOCHA [19]) and hypernetwork-based framework (i.e., FedHN [18]).
> > > However, all these methods require aggregating the model parameters from the clients, who have to apply identical model structure and size, which hinders further personalization, e.g., employing personalized model architectures for heterogeneous clients is not feasible.
> > >
> > > **Heterogeneous FL and Knowledge Distillation.** To enable heterogeneous model architectures in FL, Diao et al. [20] propose to upload a different subset of global model to the server for aggregation with the objective to produce a single global inference model. Another way of personalization is to use Knowledge Distillation (KD) in heterogeneous FL systems ([21-28]). The principle is to aggregate local soft-predictions instead of local model parameters in the server, whereby each client can update the local model to approach the averaged global predictions.
> > > As KD is independent with model structure, some literature [29,30]
> > > are proposed to take advantage of such independence to implement personalized FL with heterogeneous models at client sides. For example, Li et al. [29] propose FedMD to perform ensemble distillation for each client to learn well-personalized models. Different from FedMD that exchanging soft-predictions between the clients and the server, FedDF [30] first aggregates local model parameters for model averaging at the server side. Then, the averaged global models can be updated by performing knowledge transfer from all received (heterogeneous) client models.
> > >
> > > To sum up, most of these schemes construct an ensembling teacher by simply averaging the teachers' soft predictions, or by heuristically combining the output of the teacher models, which are far away from producing optimal combination of teachers. In our framework, KD is used in a more efficient way that the weights of the clients' soft predictions are updated together with the model parameters during every FL training iteration.
> > >
> > > ### **References**
> > > [1] Brendan McMahan, Eider Moore, Daniel Ramage, Seth Hampson, and Blaise Agüera y Arcas. Communication-efficient learning of deep networks from decentralized data. AISTATS, 2017.
> > > [2] Kangkang  Wang,  Rajiv  Mathews,  Chloé  Kiddon,  Hubert  Eichner,  Françoise  Beaufays, and  Daniel  Ramage.   Federated evaluation of on-device personalization. arXiv  2019.
> > > [3] Johannes Schneider and Michail Vlachos.  Personalization of deep learning.arXiv 2019.
> > > [4] Manoj Ghuhan Arivazhagan, Vinay Aggarwal, Aaditya Kumar Singh, and Sunav Choudhary. Federated learning with personalization layers.arXiv 2019.
> > > [5] Tao Yu, Eugene Bagdasaryan, and Vitaly Shmatikov.  Salvaging federated learning by local adaptation. arXiv 2020.
> > > [6] Canh T. Dinh, Nguyen H. Tran, and Tuan Dung Nguyen. Personalized federated learning with moreau envelopes. NeurIPS, 2020.
> > > [7] Filip Hanzely and Peter Richtárik. Federated learning of a mixture of global and local models. arXiv 2020.
> > > [8] Filip Hanzely, Slavomír Hanzely, Samuel Horváth, and Peter Richtárik.  Lower bounds and optimal algorithms for personalized federated learning. NeurIPS, 2020.
> > > [9] Tian Li, Shengyuan Hu, Ahmad Beirami, and Virginia Smith. Ditto: Fair and robust federated earning through personalization.  ICML, 2021.
> > > [10] Yishay Mansour, Mehryar Mohri, Jae Ro, and Ananda Theertha Suresh. Three approaches for personalization with applications to federated learning.arXiv 2020.
> > > [11] Yuyang Deng, Mohammad Mahdi Kamani, and Mehrdad Mahdavi.  Adaptive personalized federated learning.arXiv 2020.
> > > [12] Matthias Reisser, Christos Louizos, Efstratios Gavves, and Max Welling. Federated mixture of191experts.arXiv preprint arXiv:2107.06724, 2021.192[13]Alireza Fallah, Aryan Mokhtari, and Asuman E. Ozdaglar. Personalized federated learning with theoretical guarantees: A model-agnostic meta-learning approach. NeurIPS, 2020.
> > > [14] Yihan Jiang, Jakub Koneˇcn`y, Keith Rush, and Sreeram Kannan. Improving federated learning personalization via model agnostic meta learning.arXiv 2019.
> > > [15] Yutao Huang, Lingyang Chu, Zirui Zhou, Lanjun Wang, Jiangchuan Liu, Jian Pei, and Yong Zhang. Personalized cross-silo federated learning on non-iid data. AAAI, 2021.
> > > [16] Avishek Ghosh, Jichan Chung, Dong Yin, and Kannan Ramchandran. An efficient framework for clustered federated learning. NeurIPS, 2020.
> > > [17] Michael Zhang, Karan Sapra, Sanja Fidler, Serena Yeung, and Jose M. Alvarez. Personalized federated learning with first order model optimization.  ICLR 2021,
> > > [18] Aviv Shamsian, Aviv Navon, Ethan Fetaya, and Gal Chechik. Personalized federated learning using hypernetworks.   ICML 2021.
> > > [19] Virginia Smith, Chao-Kai Chiang, Maziar Sanjabi, and Ameet S. Talwalkar. Federated multi-task learning. NeurIPS, 2017.
> > > [20] Enmao Diao, Jie Ding, and Vahid Tarokh. Heterofl: Computation and communication efficient federated learning for heterogeneous clients.  ICLR 2021.
> > > [21] Eunjeong Jeong, Seungeun Oh, Hyesung Kim, Jihong Park, Mehdi Bennis, and Seong-Lyun Kim. Communication-efficient on-device machine learning: Federated distillation and augmentation under non-iid private data.arXiv 2018.
> > > [22] Hongyan Chang, Virat Shejwalkar, Reza Shokri, and Amir Houmansadr.   Cronus:  Robust and heterogeneous collaborative learning with black-box knowledge transfer.arXiv 2019.
> > > [23] Sohei  Itahara,  Takayuki  Nishio,  Yusuke  Koda,  Masahiro  Morikura,  and  Koji  Yamamoto. Distillation-based semi-supervised federated learning for communication-efficient collaborative training with non-iid private data.arXiv 2020.
> > > [24] Hong-You Chen and Wei-Lun Chao. Fedbe: Making bayesian model ensemble applicable to federated learning. ICLR 2021.
> > > [25] Yanlin Zhou, George Pu, Xiyao Ma, Xiaolin Li, and Dapeng Wu. Distilled one-shot federated learning.ariXiv 2020.
> > > [26] Lichao Sun and Lingjuan Lyu. Federated model distillation with noise-free differential privacy. IJCAI 2021.
> > > [27] Neel  Guha,  Ameet  Talwalkar,  and  Virginia  Smith.   One-shot  federated  learning.arXiv 2019.
> > > [28] Chaoyang He, Salman Avestimehr, and Murali Annavaram. Group knowledge transfer: Collaborative training of large cnns on the edge.arXiv 2020.
> > > [29] Daliang Li and Junpu Wang. FedMD: Heterogenous federated learning via model distillation. arXiv 2019.
> > > [30] Tao Lin,  Lingjing Kong,  Sebastian U. Stich,  and Martin Jaggi.   Ensemble distillation for robust model fusion in federated learning. NeurIPS 2020.

---

> > > > ### Comment · Reviewer_21U2 · 2021-08-31
> > > > **Feedback**
> > > >
> > > > I thank the authors for the detailed response. I'm glad that the authors took my comments seriously and made tremendous efforts in rebuttal. I have increased the score to "6".
> > > >
> > > > I also want to apologize that I missed the model average part in [16]. I initially thought that "ensemble distillation" itself can be done simply by training a model using the ensemble prediction as pseudo labels for the unlabeled data, but the model initialization does need model average in [16]. This also helps me understand footnote 2.
> > > >
> > > > Finally, I have three suggestions for the authors for their final version.
> > > > 1) Please do include the new related work and modify the introduction accordingly. Again, (a) dealing with heterogeneous systems (b) using knowledge distillation (KD) in FL (c) creating personalized global models/regularizers are all not new in the literature, **but the way the authors combine them is novel**, in which each individual component cannot be used alone for heterogeneous systems. To be clear, I don't think this is a weakness but a strength. I'm a fan of solving a new problem by a novel combination of existing solutions, rather than proposing an entirely new method (which sometimes is not necessary). Please do clearly stating how the proposed approach is related/different from these existing works. This will prevent giving readers (or even reviewers) the wrong impression that all of these components are new.
> > > >
> > > > 2) For the experiments on the homogenous systems, please do emphasize that the proposed method leverages unlabeled data. Please do include the new last table, which shows that with very few unlabeled data, the performance might degrade. But with around 500 examples (which is usually manageable), the performance/improvement becomes stable.
> > > >
> > > > 3) Please do incorporate some of my other comments and the authors' responses. For example, the way the proposed method was introduced/formulated could be improved. Some of my concerns/confusion result from some descriptions with insufficient details (like footnote 2). Also, please include the algorithm for the homogenous system.

---

> > > > > ### Author Response · Authors · 2021-09-09
> > > > > **Thanks for Reviewer 21U2's Comments and Suggestions**
> > > > >
> > > > > Thanks for your kind comments and suggestions!
> > > > >
> > > > > We’ll try our best to improve our paper in the final version.

---

> > > ### Author Response · Authors · 2021-08-28
> > > **Additional Response to Reviewer 21U2 (Part 2)**
> > >
> > > As for "Question 4", we will answer it from the following aspects:
> > >
> > > ①  We use the same experimental setting as FedDF [16] for heterogeneous FL systems, where some clients have the same model architecture. In this case, FedDF does average the local models and will obtain multiple global models at the server at the end of each communication round. So, footnote 2 of the current manuscript "we omit the parameters aggregation phase of FedDF in our experiments." is correct. Despite this fact, we realize that modifying the original FedDF is unreasonable. In the new Table we submitted several days ago, the baseline FedDF [16] means the original one, which corresponding to the above **Algorithm 3**.
> > >
> > > ②  I think there are technical/performance differences between exchanging the predictions and models for [16]. Firstly, the public data is stored in the server and cannot be downloaded to the local clients. The only way to calculate predictions is to upload local models to the server and then leverage the public dataset.
> > > Secondly, if we consider the same heterogeneous FL system settings as FedDF [16] (where some clients have the same model architecture), the local models should be uploaded to the server to perform model averaging.
> > >
> > > ③ The exact details of [16] in Table 1 of the paper can refer to above **Algorithm 3**. Both the new FedDF and pFedDF in the new table are based on exchanging models. To make it more clear, we have added a table to show the main points of each baseline:
> > >
> > > | Method | What to exchange? | Description |
> > > | :-----| :----: | :----|
> > > | FedMD [13] | Predictions | Each client first trains the local model on its private dataset, then uploads the predictions on a shared public dataset to the server. The server computes a globally averaged prediction and distributes it to all clients. Each client then trains its model to approach the averaged prediction. |
> > > | FedDF [16]| Model parameters | Each client first trains the local model on its private dataset, then uploads the models to the server to perform model averaging and knowledge distillation.  The server distributes the new fused global models to the corresponding clients. |
> > > |pFedDF| Model parameters | Using the fused prototype models at the last round of FedDF as the pFedD's initial models. Each client fine-tunes it on the local private dataset with several epochs. |
> > > |KT-pFL | Predictions | Each client first trains the local model on its private dataset, then uploads the predictions on a shared public dataset to the server. The server computes the personalized prediction for each client via a co-knowledge efficient matrix. Each client then trains its model to approach the personalized prediction. |
> > > | | | |
> > >
> > > ④ Besides, we have checked the pFedDF results and rerun the whole experiment on heterogeneous FL systems.
> > > In our experiments, we have already taken the trained fused prototype model for each client at the last round of FedDF as the pFedD's model and fine-tuned it with several epochs on local private data. The updated experimental results are shown below:
> > >
> > > | Method | EMNIST |  | FashionMNIST| | CIFAR-10 | |
> > > | :-----| :----: | :----: |  :----: |  :----: |  :----:  |  :----: |
> > > | | Non-IID_1 | Non-IID_2 | Non-IID_1| Non-IID_2|  Non-IID_1| Non-IID_2|
> > > | fedMD [13] | 71.37 | 70.15 | 68.02| 66.10| 39.14| 35.56|
> > > | FedDF [16] | 71.92|72.79 |70.32| 70.83| 40.19| 37.81|
> > > | pFedDF | 78.59| 75.72| 72.69| 73.45| 42.83| 39.54|
> > > |Sim-pFD| 82.72|74.42|72.82|72.71|42.50|39.90|
> > > |TopK-pFD|83.00| 73.91|72.24|73.39|42.58|39.34|
> > > |KT-pFL|84.65|76.30|75.48|75.60|43.82|41.04|
> > > ||||||||
> > >
> > >
> > >
> > > **Question 5**:
> > > For Q8, I have a similar question as Q7. Is there any technical or performance difference for the proposed method by exchanging the predictions or the models? I just realize that there seems to be no algorithm (like Algorithm 1) for the proposed method with exchanging model parameters. Can the authors clarify the algorithm and what the difference is from Algorithm 1? If exchanging the predictions or models has no fundamental difference for the proposed methods, I think the authors do not need to specifically mention Line 249-252, which indeed causes confusion. But if there is a difference, I'd indeed want to see the results based on exchanging model parameters to make the proposed method more consistent.
> > >
> > > **Answer 5**:
> > > In homogeneous FL systems, both baselines exchange model parameters between the clients and server. Therefore, to conduct a fair comparison with baselines, we modified the original KT-pFL (Algorithm 1 in our paper) into a new version. More specifically, the new KT-pFL exchanges model parameters and obtains a personalized model for each client by a linear combination of all local models using a parameterized manner. In this case, there are some differences between the two versions, but all use a parameterized manner to update the knowledge coefficient matrix.
> > >
> > > **Question 6**:
> > > Besides, I do want to see the results based on fine-tuning. More specifically, can the authors report the results of FedAvg --- but instead of using its global models, using its local models. I respectfully do not agree that "Per-Fedavg [9] can represent the FedAvg after local fine-tuning" since [9] only updates the global model for a few steps, not like the local models of FedAvg (and [17, k]) which are updated with local data for epochs. I also want the authors to compare algorithms with public data here. As mentioned, the reason is that "Third, the proposed method has an obvious advantage --- the public data." Alternatively, the authors can extend the experiments in the right part of Table 4 by considering even fewer public data. I believe at a certain point, the results will degrade. To clarify, it is not a bad thing, but it will be great to let the reader know the benefit of the public data. For example, if there are only 10 public data points, what will be the result?
> > >
> > > **Answer 6**:
> > > Thanks for your valuable comments. The reason why we mentioned that  "Per-Fedavg [9] can represent the FedAvg after local fine-tuning" is due to we can take the well-trained global model (i.e., via Per-Fedavg) as the initial model for each client, then the initial models are updated with local data for epochs to obtain personalized models.  To eliminate the confusion, we have added new experiments on "FedAvg + local fine-tuning" based on reference [17,k]. The detailed results can be seen below:
> > >
> > > | Method | EMNIST| FashionMNIST| CIFAR-10 |
> > > | :-----| :----: |  :----: |  :----: |
> > > |Fedavg [1]| 85.86| 80.69| 51.94|
> > > |Per-Fedavg [9]| 86.16| 85.91| 55.61|
> > > |Fedavg-Local FT [k] |86.59| 90.17| 42.88|
> > > |pFedMe [6]| 84.26| 90.97| 62.79|
> > > |FedAMP [8]| 88.51| 90.79| 60.75|
> > > | FedFomo [g]| 92.58| 88.33| 63.11|
> > > |pFedHN [m]| 94.27| 92.22| 65.31|
> > > |KT-pFL| 94.40| 93.93| 66.96|
> > > | | | | |
> > >
> > >
> > > [k] Salvaging Federated Learning by Local Adaptation, arXiv 2020
> > >
> > > Besides, we have compared the performance of our proposed methods on Fashion-MNIST with different public datasets (i.e., Task: Fashion\_MNIST; Unlabeled Open Dataset: divided from Fashion\_MNIST):
> > >
> > > | Public dataset | Test Accuracy |
> > > | :-----| :----: |
> > > |MNIST| 89.14 ($\pm$ 0.15)|
> > > |EMNIST| 88.76 ($\pm$ 0.21)|
> > > |Unlabeled Open Dataset|89.03 ($\pm$ 0.11)|
> > > | | |
> > >
> > >
> > > From the experimental results, we can observe that different settings of the public dataset have little effect on the training performance.
> > >
> > > The effect of the public data size on our proposed methods can be seen below:
> > >
> > > | Dataset |  | #Size of  | public | data | $\|\mathbb{D}_r\|$ | |
> > > | :-----| :----: | :----: | :----: | :----: | :----: | :----: |
> > > | | 10| 100 | 500 | 1000| 3000| 5000|
> > > |EMNIST (%)| 56.19| 73.44| 91.25| 91.47| 91.66| 91.32|
> > > |Fashion_MNIST (%)| 50.80| 67.43| 88.08| 89.40| 89.50| 89.14|
> > > |CIFAR-10 (%)|39.08|47.70|58.93|58.79|58.91|58.40|
> > > | | | | | | | |

---

> > > ### Author Response · Authors · 2021-08-28
> > > **Additional Response to Reviewer 21U2 (Part 1)**
> > >
> > > **Question 1** :
> > > For A1-A3, the authors seem to explain that they didn't cite/discuss those related works because they did not work on the exact setting. I respectfully do not agree with the reason. In my opinion, for a paper to be accepted, the authors should cite those papers and provide a clear discussion of why they are not applicable/feasible to the proposed setting --- which can then be seen as the motivation. As I mentioned, without citing and discussing them in the first place, a reader (or even a reviewer) may get the wrong impression that (a) dealing with heterogeneous systems (b) using KD in FL (c) creating personalized global model/regularizer are all new to the FL community. Instead, each of them has been explored earlier.
> > >
> > > **Answer 1** :
> > > We thank the reviewer for the timely feedback and valuable comments.  Sorry for the imperfect explanation on "**Related Work**". After carefully considering the suggestions, we have re-organized the structure of the "**Related Work**".
> > > A new version of "**Related Work**" has been attached at the end of the response letter.
> > > In the new version, a discussion for the existing pFL methods on homogeneous models is given first, then the related FL works on heterogeneous models are explained. Finally, we discuss the KD-based personalization frameworks in FL. By listing all mentioned literature, we emphasize the main differences between existing works and our proposed pFL method.
> > >
> > > **Question 2**:
> > > Also for A2, the authors wrote "Strictly speaking, [13] is not designed for personalized FL, but just for handling heterogeneous FL as [13] produces a single global soft prediction for each client." I respectfully do not agree. In Algorithm 1 of [13], their outputs are exactly personalized models $f_k$ for each client. The fact that [13] produces a single public prediction for KD doesn't conflict with the fact that [13] is a personalized algorithm since $f_k$ will be updated locally with local data. Note that, many existing pFL algorithms also create a single regularizer, like [6]. I don't think creating personalized regularization or personalized targets of KD is necessary for a method to be called a personalized FL algorithm.
> > >
> > > **Answer 2**:
> > > We agree with the reviewer that [13] enables federated learning among multiple clients with independently designed models. Therefore, we would revise the statements of [13] in our paper. The “**Related Work**“ has been re-organized and a new version has been attached at the end of the response.
> > >
> > > **Question 3**:
> > > For Q5, I'm indeed asking what public data are used for each experiment. I don't think the authors use the GAN-generated data. Do the authors save some data of the same dataset, like a portion of CIFAR-10, as the public data for the CIFAR-10 experiment? I may miss some details, but can the authors point me to the lines for these details?
> > >
> > > **Answer 3**:
> > > Sorry for the insufficient response for Q5, we do miss the description about what public data are used for each task in our experiment.  Actually, we use MNIST as public data to perform the distillation when using EMNIST and Fashion-MNIST as the private datasets; use CIFAR-100 as public data to perform the distillation when using CIFAR-10 as the private dataset. In our experiments, we do not save any private data at the server. Hope this answer can address your concerns.
> > >
> > > **Question 4**:
> > > For Q7, my concern is that [16] can indeed be applied in a heterogeneous system setting --- it does not average the local models, and footnote 2 of the authors' current manuscript seems to be wrong. I understand the "fair comparison" purpose by the authors, but I do want to ask if there is any technical/performance difference between exchanging the predictions or the models for [16]. I appreciate the authors' response in A7, especially with extra experiments. But, some of my questions are not addressed --- specifically, I'd like to know what the exact details are for [16] in Table 1 of the paper. (Please see my original question.) The numbers there are not in the newly provided table. Are the new FedDF and pFedDF in the new table based on exchanging models or predictions? Also, I have doubts about the pFedDF results --- in my experience and several pFL algorithms based on fine-tuning, the results will be significantly better than FedDF. More specifically, can the authors simply take the trained local models for each client at the last round of FedDF as the pFedDF models, instead of taking the final FedDF global model and fine-tuning it with just a few steps?
> > >
> > > **Answer 4**:
> > > Before answering "Question 4", we would like to explain the details of the FedDF [16]. The main point of FedDF is to aggregate knowledge from all received (heterogeneous) client models, and the distillation is performed on the server-side. In [16], the authors assume that there are some clients holding on same neural architecture in the heterogeneous FL system (refer to the statement "*We assume the system contains $p$ distinct model prototypes that potentially differ in neural architecture, structure and numerical precision. [16]*" ). After each client trains the model on its private dataset and uploads the model to the server,  the clients who have the same model architecture will perform model averaging at the server-side to obtain multiple fused global models. Then the fused global models will be distributed to the corresponding clients. Note that only when any of the two models are different, FedDF does not average the local models. To further show the exact details of FedDF [16], we list Algorithm 3 in [16] for heterogeneous system settings.
> > >
> > >
> > > **____________________________________________________________________________________________________________________________________________________________________**
> > >
> > > **Algorithm 3**: Illustration of **FedDF** for heterogeneous FL systems. The $K$ clients are indexed by $k$, and $n_k$ indicates the number of data points for the $k$-th client. The number of communication rounds is $T$, and $C$ controls the client participation ratio per communication round. The number of total iterations used for model fusion is denoted as $N$. The distinct model prototype set
> > > $\mathcal{P}$ has $p$ model prototypes, with each initialized as $x_0^P$.
> > > **____________________________________________________________________________________________________________________________________________________________________**
> > > 1: **Procedure** SERVER
> > > 2:  　　initialize HashMap $\mathcal{M}$: map each model prototype $P$ to its weights $x_0^P$.
> > > 3:  　　initialize HashMap $\mathcal{C}$: map each client to it model prototype.
> > > 4:  　　initialize HashMap $ \widetilde{\mathcal{C}}$: map each model prototype to the associated clients.
> > > 5:  　　**for** each communication round $t = 1,...,T$ **do**
> > > 6:  　　　$\mathcal{S}_t$  $\leftarrow$ a random subset ($C$ fraction) of the $K$ clients
> > > 7:  　　　**for** each client $k \in \mathcal{S}_t$ **in parallel do**
> > > 8:  　　　　$\hat{\mathbf{x}}_{t}^k \leftarrow \text{Local-ClientUpdate}(k, \mathcal{M}[\mathcal{C}[k]])$
> > > 9:  　　　**for** each prototype $P \in \mathcal{P}$  **in parallel do**
> > > 10:  　　　　initialize the client set $\mathcal{S}_t^P$ with model prototype $P$, where $\mathcal{S}_t^P \leftarrow \widetilde{\mathcal{C}}[P] \cap \mathcal{S}_t$
> > > 11:  　　　　initialize for model fusion $x_\{t,0\}^P \leftarrow \sum_\{k \in \mathcal{S}_t^P\} \frac{n_k}{\sum_\{k \in \mathcal{S}_t^P\}   n_k} \hat{x}_t^k$
> > > 12:  　　　**for** $j$ in $\\{1,...,N\\}$  **do**
> > > 13:  　　　　sample $\mathbf{d}$, from e.g. (1) an unlabeled dataset, (2) a generator
> > > 14:  　　　　use ensemble of $\\{\hat{x}_t^k\\}_\{k \in \mathcal{S}_t\} $ to update server student $x_\{t,j\}^P$ through AVGLOGITS
> > > 15:   　　　$\mathcal{M}[P] \leftarrow x_\{t,N\}^P$
> > > 16:  　**return** $\mathcal{M}$
> > > **______________________________________________________________________________________________________________________________________**

---

> ### Author Response · Authors · 2021-08-10
> **Response to Reviewer 21U2 (Part 1)**
>
> We thank the reviewer for the constructive comments. We have tried to address all your comments and suggestions. We would like to highlight that, in our response, we have answered all questions from the following three aspects:
> 1. We have explained the writing structure of the related work and provided a detailed comparison with above mentioned existing works [a-k], through which we have restated the novelty of our proposed training framework. (please refer to response **A1**-**A4**)
> 2. Moreover, we have explained the experimental design and the baseline selection. (please refer to response **A5**-**A8**)
> 3. We have added some new baselines and conducted additional experiments to further demonstrate the efficiency of our proposed method. (please refer to response  **A7**-**A8**)
>
> The detailed response for each question can be seen below:
>
> **1. Response to Question 1-4**
>
> **Q1**: There have been a few FL works on heterogeneous systems, e.g., [a, b]. Specifically, [b] has a title similar to the proposed method. Both papers are not cited. Also, there is no description in the related work on existing solutions to heterogeneous systems (e.g., [16]), giving a wrong impression that the proposed method is the first to handle this situation.
>
> [a] HeteroFL: Computation and Communication Efficient Federated Learning for Heterogeneous Clients, ICLR 2021
>
> [b] Group Knowledge Transfer: Federated Learning of Large CNNs at the Edge, NeurIPS 2020
>
> **A1**: Since we focus on improving the performance of personalized FL both on heterogeneous and homogeneous systems, we place more emphasis on discussing the limitations of existing pFL methods.
> However, none of the above works (i.e., [a,b]) consider personalization during the training process.
> Specifically, [b] aims to reduce communication overhead between the clients and server. In [a], each client uploads a different subset of global models to the server for aggregation, and the objective is to produce a single global model.
> Hence, we do not cite these works(i.e., [a, b]) as they are only related to heterogeneous systems.
> In term of [16] (i.e., FedDF), it is the first top conference paper that employs KD in FL, but do not consider the personalization of FL.
> So we do not discuss it in related work,
> and just mention it using one sentence in the introduction.
>
> **Q2**: Knowledge distillation (KD) is one promising way to handle system heterogeneity. However, instead of discussing how KD has been used in FL in the related work (e.g., [12-16]), the authors just provided a general overview of KD. Specifically, [13] is on personalized FL but without any discussion. Moreover, several other FL works using KD are not cited and discussed [c-f].
>
> [c] FedBE: Making Bayesian Model Ensemble Applicable to Federated Learning, ICLR 2021
>
> [d] Distilled one-shot federated learning, arXiv 2020
>
> [e] Federated Model Distillation with Noise-Free Differential Privacy, arXiv 2020
>
> [f] One-Shot Federated Learning, arXiv 2019
>
> **A2**: The principal reason why we do not discuss these KD-based work (e.g., [12-16]) in related work is that the main idea can be summarized by one sentence we mentioned in Line 40-42 “… knowledge can be transferred from a neural network to another via exchanging soft predictions instead of using the whole model parameters…”.
> Strictly speaking, [13] is not designed for personalized FL, but just for handling heterogeneous FL as [13] produces a single global soft prediction for each client.
> Moreover, although several FL works also use KD (e.g., [c-f]), none of them focus on personalization and can be applied to personalized FL, for example, [d] focuses on training a global model via distilled data, [c] proposes a novel model aggregation scheme in the server using Bayesian inference, which is similar to [16], [e] focuses on filling in the gaps between privacy budget and model performance in KD-based FL system. [f] leverages KD to reduce the size of the ensemble global model.
>
> **Q3**: Another important component is personalized knowledge transfer. It seems that [8, 24] share similar ideas to the proposed method, but the authors didn’t provide a detailed discussion. Some other similar works also derive personalized global models or regularizers [g, h].
>
> [g] Personalized federated learning with first order model optimization, ICLR 2021
>
> [h] Federated Mixture of Experts, OpenReview 2020
>
> **A3**: We clarify that there is no existing work on personalized knowledge distillation. Both [8,24] and [g,h] are pFL methods and need to be discussed in the first subsection of related work.
> Although [8,24] have been discussed in related work, we would still summarize the main differences between [8, 24], [g], [h] and our proposed KT-pFL.
> Firstly, FedAMP (i.e., [8]), FedFomo (i.e., [g]) and FedMix (i.e., [h]) still exchange model parameters among the server and clients, which may lead to huge communication overhead.
> Secondly, FedAMP and FedFomo maintain a personalized model for each client in the server in a non-parametrized manner, where the combination weight of each local model is calculated by a similarity function. FedMix holds $K(K \ll N)$ global models in the server and selects local clients with probability. Beyond that, FedFomo requires each client to download other clients’ models locally, so additional communication overhead and potential privacy leakage may not be ignored.
> On the other hand, MOCHA (i.e., [24]) has the same communication overhead as traditional pFL methods, meanwhile the computationally intensive of MOCHA in each client hinders the implementation for large-scale systems.
> Our work that parameterizes the combination weight matrix during training is notably different from all the above methods, significantly improving the performance on heterogeneous pFL.
>
> **Q4**: More specifically, the authors suddenly gave a formulation in Eq (2) without giving the design principle. I would suggest that the authors first describe an objective function for personalized FL, and then gradually lead to Eq (2) for the heterogeneous case. Also, I don’t fully understand the description in Line 146-147 for the regularization term in Eq (4).
>
> **A4**: We are sorry for the imperfect explanation on problem formulation. The original objective function for pFL can be represented as $\min_{w}\mathcal{L}(w)$, where $\mathcal{L}(w):=\sum_{n=1}^N \frac{D_n}{D} L_n (w^n)$, $w=[w^1,…,w^N]$. Then, taking heterogeneous models into consideration, we define the personalized loss function of client $n$ in Eq (2).
> The regularization term in Eq (4) is to ensure the collaboration between clients, especially for clients with less similarity. Without this term, a client with a completely different data distribution tends to set large values of the knowledge coefficient (i.e., equal to 1), and no collaboration will be conducted during training in this case.
>
> **2. Response to Question 5-6**
>
> **Q5**: There seems to be no clear description of how the public data is constructed. Does each communication round use the same public data? Also, there is no study on the size of the public data (not merely the batch size).
>
> **A5**: In our paper, the public dataset is selected before the training starts, which can be labeled dataset, unlabeled open dataset, and even the synthesized dataset generated by GAN. In our proposed training framework, the same public data will be used in each communication round. To further investigate the effect of public dataset size on training performance, we have conducted experiments on various public dataset sizes.
> The experimental results on the effect of the public dataset size during the distillation phase are shown on the right side of Table 4 (sorry for the wrong title "\# Batch size of public data ($|\xi_r|$)" in Table 4, the correct title should be "\# size of public data ($|\mathbb{D}_r|$)").
>
> **Q6**: Experimental designs. I checked the design of "each client contains all classes of samples" in the supplementary. It does not lead to a strong non-IID case. Please take a look at [c, i] and consider re-design the experiments
>
> [i] Measuring the effects of non-identical data distribution for federated visual classification, arXiv 2019
>
> **A6**: It is true that we do not apply the Dirichlet Distribution like the mentioned literature [c], and [i], but we clarify that our experimental design in Non-IID setting is also widely used in many works, such as [8], [1]. If it is possible, we would add more experiments on Non-IID data using Dirichlet Distribution in the future version.
>
> [1] Brendan McMahan, Eider Moore, Daniel Ramage, Seth Hampson, and Blaise Agüera y Arcas. Communication-efficient learning of deep networks from decentralized data, AISTATS, 2017.
>
> [8] Yutao Huang, Lingyang Chu, Zirui Zhou, Lanjun Wang, Jiangchuan Liu, Jian Pei, and Yong
> Zhang. Personalized cross-silo federated learning on non-iid data, AAAI, 2021.

---

### Official Review · Reviewer_7FPU · 2021-07-21

**Rating:** 8
**Confidence:** 5

**Summary:**

The work aims to address the problem of creating adequate personalized models in a federated learning regime. The problems stems from a general observation that global models do no generalize well in the Federated Learning (FL) scenarios. The authors propose to use the idea of knowledge distillation to properly transfer knowledge to different clients via soft prediction mechanics. It is based on the knowledge about predictions that can be obtained on a public dataset (also referred as collaborative knowledge). At the server side, the proposed model learns the corresponding knowledge transfer coefficients that align inputs from multiple clients and hence distill the knowledge. The training is performed in an alternating manner by switching between knowledge coefficients (updated at the server side) and the task-specific parameters of the model (updated at the client side). Advantages of the proposed approach are empirically demonstrated in comparison against other FL algorithms w.r.t. both accuracy and communication efficiency on several public datasets.

**Limitations And Societal Impact:**

FL methods are traditionally connected with privacy issues. I have outlined my privacy-related questions in the main review.

**Main Review:**

The text is well written and easy to read. It follows logical structure and is transparently organized. The related literature section is relevant and gives a good overview of the state of the field in this direction. The problem formulation and the corresponding derivations of the solution are mathematically sound and non-contradicting. The key concepts are properly introduced and clearly described. I find the way knowledge distillation utilized in this work very creative and novel. Considering that the FL topic draws a lot of attention from industry, the proposed approach seems to offer a very promising solution with high practical importance. It also open perspectives for further research in this direction, e.g. for finding better ways of aligning client data on a server side. I enjoyed reading this paper very much.

However, I also have several concerns and questions.

1) In the very beginning, the authors mention privacy-related issues in the FL context. The proposed scheme promises to resolve any issues as it doesn't directly transfer any private user data in any way. On the other hand, it seems that in order for this model to work properly, the distribution of data on public and private datasets should be close (ideally the same). If so, then knowing distribution on the public dataset $\mathbb{D}_r$ may potentially help revealing distribution on the private data $\mathbb{D}_n$, which in turn would allow for some better-then-random-guess attempts to deanonimize user data. I'm not sure what are the risks of that and it would be good to have authors add more details on this issue and the general risks of loosing privacy in the proposed scheme.

2) Why is the total number of samples summed over all $D_n$ is equal to the number of clients $N$. It basically means that there's only 1 data sample per client. But clients may generate/use many data samples. Am I missing something?

*Typos*:
- line 126: "to breaks the barrier"
- line 136: the $c$ variable should be in bold font, I guess
- line 147: "that larger", missing "is"

**Time Spent Reviewing:**

6

---

> ### Author Response · Authors · 2021-08-10
> **Response to Reviewer 7FPU**
>
> First, we thank the reviewer for recognizing the value of our work and for the valuable comments and suggestions, we will try to address your concerns as follows:
>
> **Q1**: In the very beginning, the authors mention privacy-related issues in the FL context. The proposed scheme promises to resolve any issues as it doesn't directly transfer any private user data in any way. On the other hand, it seems that in order for this model to work properly, the distribution of data on public and private datasets should be close (ideally the same). If so, then knowing distribution on the public dataset $\mathbb{D}_r$ may potentially help revealing distribution on the private data $\mathbb{D}_n$ which in turn would allow for some better-then-random-guess attempts to deanonimize user data. I'm not sure what are the risks of that and it would be good to have authors add more details on this issue and the general risks of loosing privacy in the proposed scheme.
>
> **A1**: We agree with the reviewer that the distribution of data on public and private dataset should be close in theory, however, knowing the distribution on public dataset only reveals the mixture distribution of all private data (ideally). Only when the private data is IID can the knowing distribution on public data represent the specific distribution on private data $\mathbb{D}_n$.  When it comes to Non-IID case, the privacy of the local distribution on $\mathbb{D}_n$  will not be revealed since the local dataset does not represent the mixture/global distribution.
>
> Secondly, many existing works have already demonstrated that the diversity of the public data does not significantly impact the performance of distillation [i.e., 16]. Therefore, it is no need to select a public dataset with the same distribution as private data. In the paper (footnote 1), we also point out that the public data can also be unlabeled open data instead of data already labeled, and our proposed training framework works well in both cases.
>
> [16] Tao Lin, Lingjing Kong, Sebastian U. Stich, and Martin Jaggi. Ensemble distillation for robust model fusion in federated learning. NeurIPS, 2020.
>
> **Q2**: Why is the total number of samples summed over all $\mathbb{D}_n$ is equal to the number of clients $N$. It basically means that there's only 1 data sample per client. But clients may generate/use many data samples. Am I missing something?
>
> **A2**: Sorry for the typo error, we use an incorrect notation to represent the total number of samples over all $\mathbb{D}_n$. It should be $D$ instead of $N$.

---

### Decision · Program_Chairs · 2021-09-27

**Decision:**

Accept (Poster)

**Comment:**

This paper proposes a novel approach for handling model architecture heterogeneity in federated learning. Theoretical analysis and numerical experiments demonstrate the efficacy of the new method. Good writing and clear structure also make the paper stand out. Based on the reviews, I recommend acceptance. Meanwhile, please make sure to incorporate the reviewers' comments and the rebuttal into the final version.